# The tuning of tuning: How adaptation influences single cell information transfer

**Fleur Zeldenrust**[1]*, **Niccolò Calcini**[2], **Xuan Yan**[3], **Ate Bijlsma**[4], **Tansu Celikel**[5]

**1** Donders Institute for Brain, Cognition, and Behaviour, Radboud University, Nijmegen - the Netherlands, **2** Maastricht Centre for Systems Biology (MaCSBio), University of Maastricht, Maastricht, The Netherlands, **3** Institute of Neuroscience, Chinese Academy of Sciences, Beijing, China, **4** Department of Population Health Sciences / Department of Biology, Universiteit Utrecht, the Netherlands, **5** School of Psychology, Georgia Institute of Technology, Atlanta - GA, United States of America

☉ These authors contributed equally to this work.
* fleur.zeldenrust@donders.ru.nl

**Data Availability Statement:** We have made the code and data freely available in an open repository (https://doi.org/10.34973/4f3k-1s63). The scripts for generating the input current used in the

## Abstract

Sensory neurons reconstruct the world from action potentials (spikes) impinging on them. To effectively transfer information about the stimulus to the next processing level, a neuron needs to be able to adapt its working range to the properties of the stimulus. Here, we focus on the intrinsic neural properties that influence information transfer in cortical neurons and how tightly their properties need to be tuned to the stimulus statistics for them to be effective. We start by measuring the intrinsic information encoding properties of putative excitatory and inhibitory neurons in L2/3 of the mouse barrel cortex. Excitatory neurons show high thresholds and strong adaptation, making them fire sparsely and resulting in a strong compression of information, whereas inhibitory neurons that favour fast spiking transfer more information. Next, we turn to computational modelling and ask how two properties influence information transfer: 1) spike-frequency adaptation and 2) the shape of the IV-curve. We find that a subthreshold (but not threshold) adaptation, the 'h-current', and a properly tuned leak conductance can increase the information transfer of a neuron, whereas threshold adaptation can increase its working range. Finally, we verify the effect of the IV-curve slope in our experimental recordings and show that excitatory neurons form a more heterogeneous population than inhibitory neurons. These relationships between intrinsic neural features and neural coding that had not been quantified before will aid computational, theoretical and systems neuroscientists in understanding how neuronal populations can alter their coding properties, such as through the impact of neuromodulators. Why the variability of intrinsic properties of excitatory neurons is larger than that of inhibitory ones is an exciting question, for which future research is needed.

## Author summary

Intracellular information transfer from synaptic input to output spike train is necessarily lossy. Here, we explicitly measure the mutual information between a neuron's input and spike output and show that information transfer is more lossy and heterogeneous for

experiments and simulations can be found in this GitHub repository: https://github.com/DepartmentofNeurophysiology/Analysis-tools-for-electrophysiological-somatosensory-cortex-databank.

**Funding:** This work was supported by grants from the European Commission (Horizon2020, nr. 918 660328), European Regional Development Fund (MIND, nr. 122035) and the Netherlands Organisation for Scientific Research (NWO-ALW Open Competition, nr. 920 824.14.022) to TC and by the Netherlands Organisation for Scientific Research (NWO 921 Veni Research Grant, nr. 863.150.25) to FZ. The funders had no role in study design, data collection and analysis, decision to publish, or preparation of the manuscript.

**Competing interests:** The authors have declared that no competing interests exist.

excitatory than for inhibitory neurons. By using computational modelling we show that the shape of the input-output curve as well as how fast a neuron adapts to its input collectively determine the rate of information loss. These insights will help both experimentalists and modellers in designing and simulating experiments that investigate how network coding properties can adapt to the environment, for instance through the effects of neuromodulators.

## Introduction

Perception and other brain functions require information transmission and signal transformation at each processing step. Specifically, for perception, stimuli that impinge on sensory receptors are transferred via the brain stem and thalamus to cortical networks: each of these processing steps results in information transfer and compression, due to intracellular information transfer from synaptic input current to spike train. The spike train of a single neuron though, can contain only a limited amount of information about an incoming stimulus [1]. However, the working range of a neuron is typically limited, more limited than the range of inputs a neuron might receive. A neuron's ability to adapt its working range to the properties of the stimulus is crucial for its ability to transfer information about the stimulus to the next processing level [2–5]. For example, if the input amplitude is too low, a neuron that cannot adapt will not respond, whereas when the input amplitude is too large, a neuron that cannot adapt will enter depolarization block or its output firing rate will be saturated, both resulting in a neuron that does not respond adequately to changes in input and hence in a neuron that does not transfer information. Therefore, neurons need to continually adapt their working range (i.e. their excitability) in order to fit the dynamic range of the input. They can do this by reducing synaptic strength [6, 7] or by shifting (gain shift) or widening (gain modulation) their intrinsic excitability [8–10], This changing of the intrinsic input-output curves happens on different timescales: from fast (spike frequency adaptation [11]) to slow (homeostatic scaling, for reviews see [7, 12]). The dynamics of such adaptation mechanisms impact the effectiveness of the adaptation in relation to the stimulus dynamics: if the adaptation is too fast (relative to the input statistics), it has no practical effect, but if it is too slow, it is constantly saturated and has no dynamic effects. Here, we focus on the relatively fast adaptive changes in intrinsic excitability and ask how such mechanisms influence information transfer in cortical neurons and how tight their properties need to be tuned to the stimulus statistics for them to have an effect.

We start by measuring the intrinsic information encoding properties of putative excitatory (regular-spiking) and inhibitory (fast-spiking) neurons in L2/3 of the mouse barrel cortex. We measure the effects of several intrinsic neural characteristics on the information transfer from input current to output spike train, using a combination of ex-vivo experiments [13, 14] and computational modelling. We aim to unravel how both the threshold behaviour and the I-V curve shape of excitatory and inhibitory neurons affect information transfer, using a recently developed method to estimate the mutual information between input and output in an ex-vivo setup [15]. This method has several advantages: instead of the traditionally long ($\sim$ 1 hour) experiments that are needed to obtain a single mutual information estimate [16–20], this method needs only about 5 minutes of recording to obtain an information transfer estimate. Moreover, the properties of the input current can be adapted to fit different cell type properties, and it has an optimal observer model so that the measured information transfer can be compared with the 'optimal' Bayesian Neuron information transfer [21]. Using this method,

we can simultaneously measure the information transfer from input current to output spike train and assess intrinsic cell properties, thereby showing how intrinsic cell properties correlate with information transfer. In particular, putative excitatory neurons show high thresholds and strong adaptation, making them fire sparsely and resulting in a strong compression of information between input and output. Their intrinsic properties are quite heterogeneous, showing a large variability. Putative inhibitory neurons on the other hand have intrinsic properties that favour higher firing rates, corresponding to a higher information transmission rate. Their response properties are more stereotypical than those of excitatory neurons.

The Bayesian neuron that is optimal for this task has two properties that distinguish it from a standard leaky integrate-and-fire model: 1) spike-frequency adaptation and 2) a non-linear I-V curve that results amongst others in the suppression of hyperpolarization. To untangle how these mechanisms influence information transfer, we turn to computational modelling. Firstly, we use an exponential integrate-and-fire (expIF) model with two types of adaptation: subthreshold adaptation [22, 23] and threshold adaptation [24] and research the effects of these two types of adaptation on the information transfer in the aforementioned mutual information protocol. We find that subthreshold adaptation increases the information transfer if tuned well, whereas threshold adaptation increases the working range of the neuron over a broad range of parameters. So despite the fact that at first glance these two forms of adaptation appear to serve a similar purpose (i.e. reducing the firing rate of a neuron for strong stimuli), it turns out that their effects are quite different. Secondly, we assess the effects of changing the shape of the I-V curve (the right-hand-side of the membrane voltage equation). We model the effects of suppression of hyperpolarization by adding an instantaneous 'h-current' to the expIF neuron, the effects of an instantaneous subthreshold potassium current, and the effects of changing the leak conductance of the neuron. We find that a well-tuned subthreshold (but not threshold) adaptation, the 'h-current', and a properly tuned leak conductance can increase the information transfer of a neuron, whereas threshold adaptation can increase its working range.

## Materials and methods

### Ethics statement

Animals used were Pval-cre and SSt-cre mice from 9 to 45 weeks kept with unlimited access to water and food, housed in a 12-hour light/dark cycle. All experimental procedures were performed according to Dutch law and approved by the Ethical Committee for Animal Experimentation of Radboud University (RU DEC) as described before (for further details, see [25, 26]). Each mouse was perfused with iced and oxygenated ($95\%O_2/5\%CO_2$) Slicing Medium (composition in mM: $108ChCl$, $3KCl$, $26NaHCO_3$, $1.25NaH_2PO_4H_2O$, 25 Glucose.$H_2O$, $1CaCl_{2.2}H_2O$, $6MgSO_{4.7}H_2O$, 3 Na-Pyruvaat) under anaesthesia with 1,5ml Isofluraan.

### Experiments

All analyzed current clamp and simulation data and the code to analyze and simulate them can be found in this repository: https://doi.org/10.34973/4f3k-1s63. The voltage clamp data are part of the dataset of da Silva Lantyer et al. (2018) [13].

**Slice electrophysiology.** The brain was covered in 2% agarose and submerged in a Slicing Medium after which it was sliced in 300 $\mu M$ thickness using a VF-300 compresstome (Precisionary Instruments LLC) and then incubated for 30 min in 37˚C artificial cerebrospinal fluid (ACSF, composition in mM: $1200NaCL$, $35KCL$, $13MgSO_{4.7}H_2O$, $25CaCl_{2.2}H_2O$, 100 Glucose.$H_2O$, $12.5NaH_2PO_4.H_2O$, $250NaHCO_3$), oxygenated ($95\%O_2/5\%CO_2$). The bath was then transferred to room temperature. Slices were allowed to accomodate to room temperature for

30 min and were kept in this bath until use. Slices were placed into the recording chamber under the microscope (Eclipse FN1, Nikon) and perfused continuously at a rate of 1 ml/min with the oxygenated ACSF at room temperature. Patch pipettes for whole-cell recordings were pulled from borosilicate glass capillaries, 1.0 mm outer diameter, 0.5mm inner diameter, on a pipette-puller (Sutter Instrument Co. Model P-2000), until an impedance of 8±2 MΩ for the tip was obtained. Pipettes were filled with a solution containing (in mM) $115 CsMeSO_3$, $20 CsCl$, $10 HEPES$, $2.5 MgCl_2$, $4 Na_2ATP$, $0.4 NaGTP$, $10 Na$ – phosphocreatine, $0.6 EGTA$, $5 QX$ – 314 (Sigma). The whole cell access was obtained after reaching the gigaohm seal and breaking the membrane. Upon entering the cell and the whole-cell mode, the membrane potential was kept fixed at -70mV, outside stimulation.

**Input current generation.** Data acquisition was performed with HEKA EPC9 amplifier controlled via HEKA's PatchMaster software (version 2.90x.2), and subsequent analysis with MatLab (Mathworks, v.2016b). Three types of experiments were performed: current clamp (CC) step-and-hold, current clamp (CC) frozen noise, and voltage clamp (VC).

The current clamp (CC) step-and-hold protocol was performed in every cell and used to distinguish between cell types, according to the firing rate and spike shape (Fig 1). The protocol consisted of clamping the neuron at a baseline current $I_{baseline}$, corresponding to the one required to keep its membrane at -70mV, and providing a 500ms long stimulus of fixed current value $I = I_{baseline} + (40pA * \text{step number})$, for a total of 10 steps, reaching a maximum current injected of $I_{baseline} + 400pA$. Between each current injection step, a 5.5s recovery window was allowed.

Information transfer was measured using the 'frozen noise' method introduced by Zeldenrust et al. [15]. To measure information transfer from input to spike train in short periods of time, instead of the long measurements needed for the traditional methods (see Introduction), the noisy input current injected into a neuron in the current clamp setting was generated as the output of a simulated Poisson neural network (SPNN) that responded to a randomly appearing and disappearing preferred stimulus or 'hidden state' (Markov process) $x$: a binary variable that can take the values of 1 (preferred stimulus present, 'on-state') and 0 (preferred stimulus absent, 'off-state', see Fig 2A). This hidden state is switched on and off according to a Markov process with rates $r_{on}$ and $r_{off}$. The advantage of using such a binary hidden state stimulus is that there is no need to reconstruct the full input current (which is high-dimensional and therefore requires long recordings), but it is sufficient to reconstruct the binary stimulus, for which less data is needed. The $N = 1000$ neurons of the SPNN fired Poisson spike trains, whose firing rates were modulated by the hidden state stimulus, so that each neuron fired with rate $q_{on}^i$ when $x = 1$, and $q_{off}^i$ when $x = 0$. These rates were drawn from a Gaussian distribution with mean $\mu_q$ (see Table 1) and standard deviation $\sigma_q = \sqrt{\frac{1}{8}\mu_q}$. Each spike was convolved with an exponential kernel with a unitary surface and a decay time of 5ms. The spike trains from different presynaptic neurons contribute to the output with weight $w_i = \log\left(\frac{q_{on}^i}{q_{off}^i}\right)$. In order to choose the parameters of this input current two considerations needed to be made:

1. The Markov process had rates $r_{on}$ and $r_{off}$, which correspond to a switching time constant $\tau_{input} = \frac{1}{r_{on}+r_{off}}$: since excitatory neurons do not fire at rates higher than a few Hz, but the inhibitory neurons show a much broader working range, up to 100 Hz (Fig 1), the information transfer could not be measured in the two types of neurons with the same switching speed of the hidden state $\tau_{input}$, but the switching speed of the hidden state needed to be adapted to the working range of the neuron type. Fortunately, the method allows for keeping the information about the hidden state in the input constant while changing $\tau_{input}$, by adjusting the mean firing rate of the SPNN $\mu_q$ (see Fig 2B). So the information transfer in

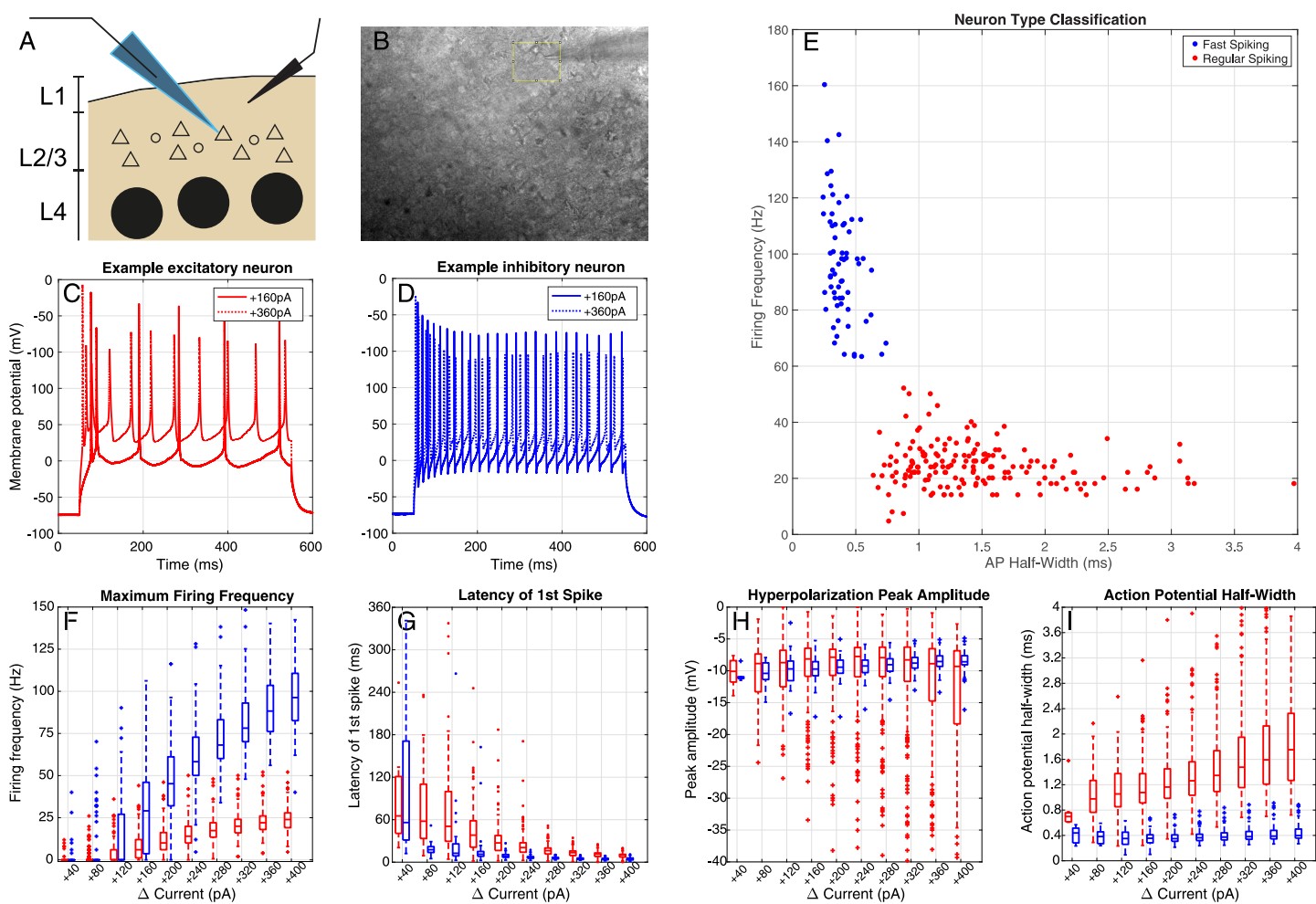

**Fig 1. Cell Classification with the CC step-and-hold protocol.** A, B: We selected neurons to record from the mouse somatosensory cortex (barrel cortex), in L2/3. Visually, the shape and size of soma were a good indicator of the cell type: smaller and roundish shapes would point towards fast-spiking neurons, while slightly larger and triangular shapes would point to regular spiking (putative excitatory) neurons. C: Example responses of an excitatory cell to a constant injected current. D: Example responses of an inhibitory cell to a constant injected current. E: Cell classification using agglomerative clustering based on the maximum firing frequency and spike width. Cell classification using agglomerative clustering based on the maximum firing frequency and spike width. This method was used to verify that cells were correctly classified as inhibitory (blue) or excitatory (red) during the experiment: in pink the cell(s) where the agglomerative clustering and the initial classification disagreed, and were excluded from the analysis (see Materials & methods). F: Maximum firing frequency distribution for incremental current injection amplitudes for inhibitory (blue) and excitatory (red) neurons. G: Same as F, but for the latency of the first spike. H: After-hyperpolarization distribution. I: Spike half-width distribution. For threshold behaviour in the current-clamp step-and-hold protocol, see S1 Fig.

the two neuron types could still be compared by choosing a time constant $\tau_{\text{input}}$ of 50 or 250 ms for inhibitory/fast spiking or excitatory/regular spiking neurons respectively, with matching values of $\mu_q$ of 0.5 Hz or 0.1 Hz respectively (see Table 1, so that the mutual information between the input current and the hidden state ($MI_I$) was about 0.3 bit (Fig 2E). This target mutual information between the input current and the hidden state was chosen so that the input current was informative about the hidden state, but not too informative.

2. The input generated by the SPNN responding to the Markov Process ($I_{\text{Markov}}(t)$) is dimensionless. Therefore, this dimensionless theoretical "input current" needed to be scaled to Ampère so that it could be injected into the neuron in a current clamp setup. Therefore, the injected current was defined as $I_{\text{injected}} = I_{\text{hold}} + I_{\text{scale}}I_{\text{Markov}}(t)$. Here, the neuron was

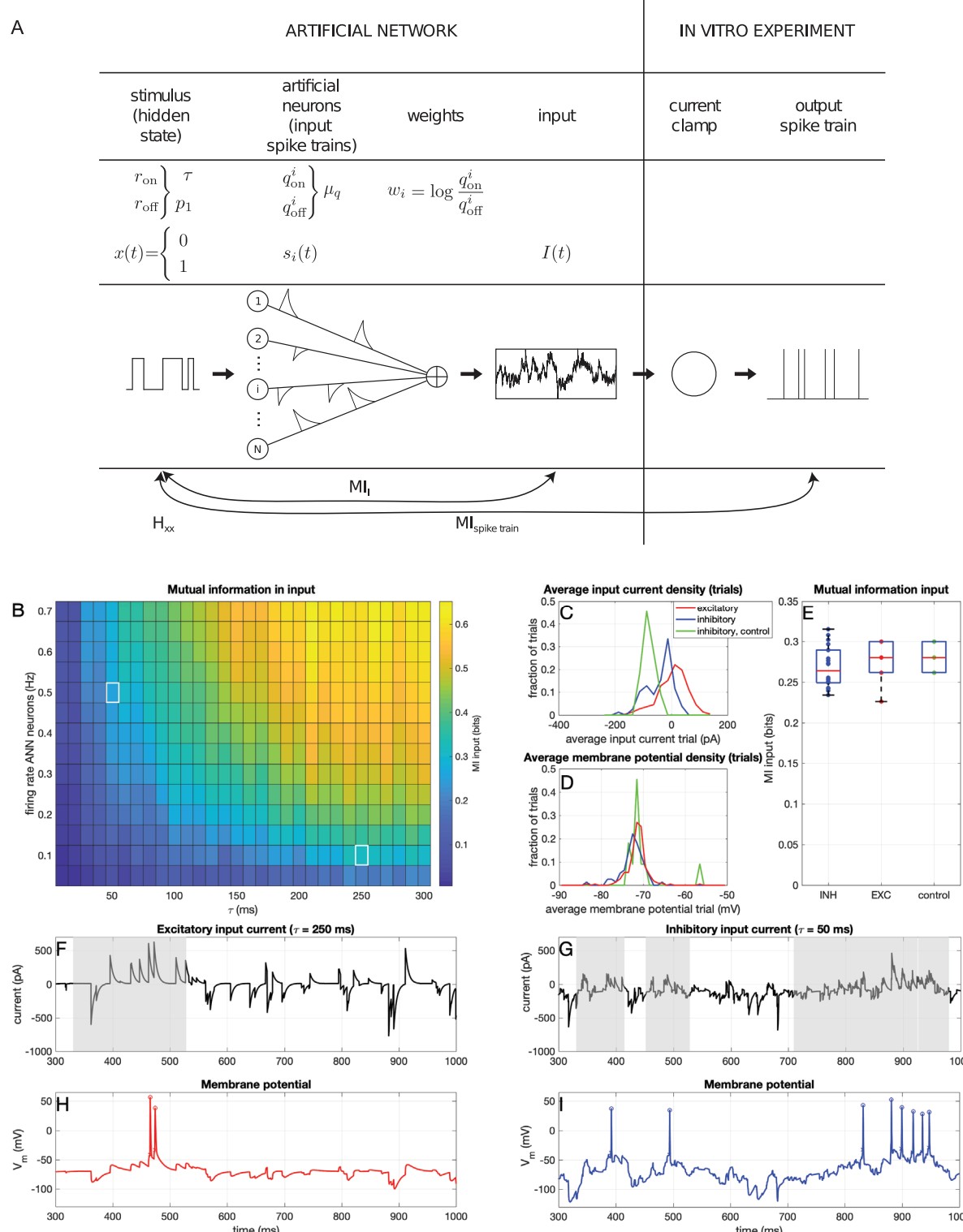

**Fig 2. Input for the CC frozen noise protocol.** A: Overview of the frozen noise method for input generation and measurement of mutual information (copied with permission from [15]) B: Mutual information between the input current and the hidden state, for different values of the switching speed of the hidden state ($\tau_{\text{input}}$) and the average firing rate of neurons in the SPNN (average over 10 trials). The white squares denote the used values for the input for the inhibitory (top left) and excitatory (bottom right) neurons. C: Average (over the trial) input current (i.e. $I_{\text{baseline}}$ and D: membrane potential for all trials. Green data points/lines denote the control experiments where the inhibitory neurons

received the input current that was otherwise given to the excitatory neurons. E: Mutual information between the hidden state and the input current, for all trials. Note that because frozen noise was used, every frozen noise trial was actually the same. Therefore, there are not many different realizations and hence not many different MI values. F: Example injected frozen noise current for an excitatory neuron. The grey shaded area corresponds to times when the hidden state was 1. G: Example injected frozen noise current for an inhibitory neuron. H: Example resulting membrane potential of an excitatory neuron. I: Example resulting membrane potential of an inhibitory neuron.

clamped a baseline current $I_{\text{baseline}}$, corresponding to the current required to keep its membrane at -70mV, and $I_{\text{scale}}$ was set at 2100 pA for excitatory and 700 pA for inhibitory cells (see Table 1). This way, the input was scaled to be at roughly the same point on each neuron's I-V curve.

The scripts for generating this current can be found in this GitHub repository: https://github.com/DepartmentofNeurophysiology/Analysis-tools-for-electrophysiological-somatosensory-cortex-databank.

In the experimental paradigm, we used a 'frozen' current injection: the exact same current (except for the overall baseline and amplitude) for all excitatory neurons and the exact same one for all inhibitory neurons. This had of 2 reasons: the first was an experimental setup constraint: the current had to be generated offline and loaded onto the setup, which would have made the experiments take a lot longer to perform if we did not use the same current each time. The second was that the experiments were in many neurons also repeated with different neuromodulators [14]. For proper comparisons between aCSF and neuromodulator trials (ie omitted or added spikes for the same input), the input current needed to be frozen. These neuromodulator trials were not included into this manuscript; only the aCSF condition was used.

## Analysis

**Cell classification.** Cells were classified using the following procedure: each cell was first stimulated with a current-clamp step-and-hold (CC-step) protocol, which was used to classify the cells on-site, during the experiment, before generating the frozen noise stimulus with parameters matching that cell group. Cells were assigned to the fast spiking (putatively inhibitory) or regular spiking (putatively excitatory) groups, on the basis of the maximum firing rate recorded during the CC-step protocol (in particular, if this was lower than $\approx 50$Hz), and the action potential shape (Fig 1). Based on this initial classification the cell received the frozen noise input current with either $\tau_{\text{input}} = 250$ ms (excitatory neurons) or $\tau_{\text{input}} = 50$ ms (inhibitory neurons). From these recordings, the maximum firing rate, the average spike-halfwidth,

**Table 1. Input parameters for the ex-vivo experiments.**

| Parameter | Excitatory cells | Inhibitory cells |
|---|---|---|
| Number of artificial neurons $N$ | 1000 | 1000 |
| Hidden state time constant $\tau_{\text{input}}$ | 250 ms ($r_{\text{on}} = 1.3$ Hz, $r_{\text{on}} = 2.7$ Hz) | 50 ms ($r_{\text{on}} = 6.7$ Hz, $r_{\text{on}} = 13.3$ Hz) |
| Average firing rate artificial neurons $\mu_q$ | 0.1 Hz | 0.5 Hz |
| Baseline input current $I_{\text{baseline}}$ | (set so the cell was at -70 mV, see Fig 2) | (set so the cell was at -70 mV, see Fig 2) |
| Amplitude input current $I_{\text{scale}}$ | 2100 pA | 700 pA |
| Analysis window size | 100 s | 20 s |
| Number of measured cells | 144 | 72 (+ 9 control) |
| Number of trials | 220 | 78 (+11 control) |

the latency to the generation of the first action potential, and the average after-hyperpolariza-tion (AHP) amplitude were extracted (Fig 1). Offline, after data collection, the initial classifica-tion was verified using an agglomerative clustering protocol (MATLAB 'clusterdata') to cluster the data into 2 groups (separated following Ward's linkage method), according to the maxi-mum firing rate and the average spike-half width (normalized to zero mean and unit standard deviation) reached during the CC-Step protocol (Fig 1E). There was a single cell where the ini-tial classification and the post-hoc classification were in disagreement (Fig 1E, pink star). We decided to keep this cell in the original (inhibitory) group due to its position between the two clusters.

**Calculation of mutual information.** The mutual information between the hidden state and the input ($MI_I$) or a spike train ($MI_{\text{spike train}}$) was estimated with the help of the hidden state $x$ (see Input current generation and Fig 1A). The method was explained in detail in Zel-denrust et al. (2017) [15] and followed derivations from Denève (2008) [21] and Lochmann and Denève (2011) [27]). Code for how to calculate the mutual information can be found in the following repository: https://github.com/DepartmentofNeurophysiology/Analysis-tools-for-electrophysiological-somatosensory-cortex-databank as well as with the data.

In short, the mutual information was calculated using the log odds ratio that the hidden state is 1, given the history of the input until now $I(t)$

$$L(t) = \log \frac{p(x = 1 | I(t))}{p(x = 0 | I(t))} \tag{1}$$

$$= \log \frac{p(x = 1 | I(t))}{1 - p(x = 1 | I(t))}. \tag{2}$$

The *estimated* log-odds ratio $\hat{L}$ that the hidden state is 1, given the history of the input until now $I(t)$ can be calculated by integrating the following differential equation (see [15, 21] for the derivation):

$$\frac{d\hat{L}}{dt} = r_{\text{on}}(1 + e^{-\hat{L}}) - r_{\text{off}}(1 + e^{\hat{L}}) + I(t) - \theta, \tag{3}$$

where $\theta = \sum_{i=1}^{N} q_{\text{on}}^i - q_{\text{off}}^i$ is the constant offset of the input, which is chosen to equal to 0 by generating the input as explained before (drawing $q_{\text{on}}^i$ and $q_{\text{off}}^i$ from a Gaussian distribution). We can use the definition of the log-odds ratio to estimate the probability that the hidden state is 1, given the history of the input until now:

$$\hat{p}(x = 1 | I(t)) = \frac{1}{1 + e^{-\hat{L}(t)}}$$

Using this estimate of the probability that the hidden state is 1 over time, we can now estimate the conditional entropy by averaging over time, where we used an ergodic argument to approximate an average over samples by an average over time.

$$\hat{H}_{xy} = -\langle x(t) \log_2 (\hat{p}(x = 1 | I(t))) + (1 - x(t)) \log_2 (1 - \hat{p}(x = 1 | I(t))) \rangle_{\text{time}}. \tag{4}$$

Because the hidden state follows a memory-less Markov process, its entropy at every moment in time equals

$$H_{xx} = -P_1 \log_2 (P_1) - (1 - P_1) \log_2 (1 - P_1). \tag{5}$$

Here, $P_1 = \frac{r_{\text{on}}}{r_{\text{on}} + r_{\text{off}}}$ is the prior probability that the hidden state equals 1. Note that in our case,

because $P_1$ is smaller than 0.5, the entropy of the hidden state is smaller than 1 ($H_x x = 0.92$ bits). With the canonical $MI = H_{xx} - H_{xy}$, the mutual information between the input and the hidden state can now be estimated. Similarly, the mutual information between a spike train and the hidden state can be estimated by integrating Eq 3 where the input $I(t)$ is now replaced by the spike train input given by

$$I_{\text{spike train}}(t) = w \cdot \rho(t),$$ (6)

where $\rho(t)$ is the spike train of the neuron and its weight $w$ is given by

$$w = \log_2 \frac{\hat{q}_{\text{on}}}{\hat{q}_{\text{off}}} = \log_2 \left( \frac{\#\text{spikes while } x = 1}{\#\text{spikes while } x = 0} \cdot \frac{\text{total duration } x = 0}{\text{total duration } x = 1} \right).$$ (7)

Parameter $\theta = \sum_{i=1}^{N} q_{\text{on}} - q_{\text{off}}$ is calculated similarly based on the observed $q_{\text{on}}$ and $q_{\text{off}}$ in the spike train.

Note that even though theoretically $MI \leq 0$, due to our approximation it can happen that our estimate of $H_{xy} > H_{xx}$, due to the integration method, and hence that we find a small negative value of the MI between a spike train and the hidden state. These are often cells that fire either at very low rates or have firing patterns that for other reasons deviate from a Poisson-like response (i.e. cells that stop firing during the experiment). However, to maintain a complete overview of the data, we decided not to discard those that have a low firing rate. However, we did exclude files with negative values in the input current (where the wrong input current was saved) and files with a vanishing firing rate.

In this manuscript, we mostly report on the unitless 'Fraction of Information' (*FI*) in the output spike train:

$$FI = \frac{MI_{\text{spike train}}}{MI_{\text{input}}}.$$ (8)

The *FI* quantifies how much information about the hidden state is transferred from the input current to the spike train, and thus quantifies which fraction of the information is kept during the spike-generating process.

**Alternative calculation of mutual information.**   For comparison, we used the following alternative method of calculating the mutual information between the hidden state and the input current, membrane potential or spike train. The results are shown in S2 Fig. When the methods are compared, the method of the previous paragraph is referred to as the 'Bayesian MI' and the method explained here as the 'binned MI'.

To calculate the mutual information between the hidden state and any given state parameter $y$ (i.e. the input current, membrane potential or instantaneous firing rate), we first divide the time series in periods between state switches (i.e. periods for which the hidden state is constant, which we will call 'states'), and average the state parameter $y$ over this 'state'. This way, we obtain an average value for $y$ for the duration of an 'up-state' and 'down-state': $\langle y \rangle$. Next, we make a distribution of all those average values of $y$ (see S2B–S2E and S2H–S2K Fig). We can then calculate the Kullback-Leibler divergence $D_{KL}$ between the distribution of average values when the hidden state is 1 and the distribution when the hidden state is 0 (see S2F, S2L, S2N and S2O Fig). In particular, we calculated the Jensen-Shannon divergence (using the code of [28]), or half the 'Jeffrey divergence' [29]. Next, we calculate the mutual information directly, by binning the two histograms (we will call the bins $b_{\langle y \rangle}$ and the value of $\langle y \rangle$ in bin $b$

$\langle y \rangle_b$) of and calculating

$$MI_{\text{binned}} = \sum_{\text{state}} \sum_{b_{\langle y \rangle}} P(\text{state}, \langle y \rangle = \langle y \rangle_b) \log_2 \frac{P(\text{state}, \langle y \rangle = \langle y \rangle_b)}{P(\text{state})P(\langle y \rangle = \langle y \rangle_b)} \qquad (9)$$

$$= \sum_{b_{\langle y \rangle}} P(x = 1, \langle y \rangle = \langle y \rangle_b) \log_2 \frac{P(x = 1, \langle y \rangle = \langle y \rangle_b)}{P(x = 1)P(\langle y \rangle = \langle y \rangle_b)} \qquad (10)$$

$$+ P(x = 0, \langle y \rangle = \langle y \rangle_b) \log_2 \frac{P(x = 0, \langle y \rangle = \langle y \rangle_b)}{P(x = 0)P(\langle y \rangle = \langle y \rangle_b)}. \qquad (11)$$

Based on S2F and S2L Fig, we used 500 bins in our calculations.

To calculate the mutual information between the hidden state and the instantaneous firing rate of the neuron, we have to make an estimate of the instantaneous firing rate based on the spike train. This is typically done by convolving the spike train with a a kernel (see [30] for a thorough discussion). We need a causal kernel, because it would not make sense to see the firing rate increase before a state switch. Therefore, we use an exponential kernel with a unit surface. Also, to make the calculation comparable to the 'Bayesian MI', we make the decay time constant of the exponential kernel equal to the time constant of the input $\tau_{\text{input}}$, so 50 ms for inhibitory neurons and 250 ms for excitatory neurons.

The two methods result in very comparable values for the mutual information between the hidden state and the spike train (see S2M Fig). Moreover, we can calculate the slope of the IV curve (S2D and S2J Fig) and determine the correlation with the mutual information (S2P Fig). This leads to the same conclusion as with the 'Bayesian MI' and the dynamic IV curve (i.e. a greater resistance $R_{\text{in}}$/ lower conductance $g$ is associated with a larger information transfer). Because the two methods result in similar conclusions, this method is not used throughout the paper and its results are only shown in the Supplementary Material for reference. However, here we will shortly discuss the advantages and disadvantages of both methods. In terms of computational resources, the 'binned MI' computational costs come from the convolution of the spike train with the kernel to obtain the instantaneous rate, and for the 'Bayesian MI' most computational costs come from the integration of the log-likelihood (Eq 3). Which one requires more computational resources depends on the parameters of the methods and properties of the spike trains. In terms of intuition, the 'binned MI' might be more intuitive to understand (a measure of how easy it is to distinguish the two values of the hidden state depending on observing average state parameter $\langle y \rangle$). However, for this method one also needs to make a few non-trivial choices (number of bins, kernel size and shape for the calculation of the instantaneous firing rate). Finally, a more fundamental difference between the two methods, is that the 'binned MI' is post-hoc, and can only be calculated after observing a large amount of states, whereas the 'Bayesian MI' is essentially an online method, that keeps an estimate of the current most likely value of the hidden state online. Which method is more appropriate will depend on the experimental/simulation setup. A final observation is that the Kullback-Leibler divergence between the histograms for the two states is larger for the membrane for the input, but smaller again for the firing rate, something we have observed before [1].

**Threshold detection.** The membrane potential threshold of each recorded spike in the Current Clamp (CC) experiments was determined from the experimentally recorded membrane potential using the method explained in [24]: in a window from 1 to 0.25 ms before each spike maximum, the earliest time in the window at which either the first derivative exceeded

18 mV/ms or the second derivative exceeded 140 mV/ms2 was designated as the threshold-time, and the threshold value was determined as the corresponding membrane potential of that time point.

**ROC curves.**   A Receiver Operator Characteristic (ROC) curve shows how well a system can be classified into two binary classes by comparing the number of correctly detected positives or 'hits' to the number of false positives or 'false alarms' depending on a threshold parameter. Here, we assumed every trial had its own threshold, and we defined a 'hit' as a period during which the hidden state was 1, in which at least 1 action potential was fired, and a 'miss' as a period during which the hidden state was 1, in which no action potentials were fired. Similarly, we defined a 'false alarm' as a period during which the hidden state was 0, in which at least 1 action potential was fired, and a 'correct reject' as a period during which the hidden state was 0, in which no action potentials were fired. So each period in which the hidden state was 1, was either defined as a 'hit' or a 'miss', and each period in which the hidden state was 0, was either defined as a 'false alarm' or a 'correct reject'. The total number of hits was divided by the total number of periods during which the hidden state was 1, which resulted in the fraction of hits $0 \leq f_h \leq 1$. Similarly, the fraction of 'misses', 'false alarms', and 'correct rejects' were defined as the fraction of periods during which the hidden state was 1 but no spike was fired, the fraction of periods during which the hidden state was 0, but a spike was fired and the fraction of periods during which the hidden state was 0, and no spike was fired, respectively. We calculated the fractions of hits, misses, false alarms, and correct rejects for each spike train, as well as for a corresponding Poisson spike train of the same length and with the same number of spikes. Note that for these Poisson spike trains, the hit fraction is actually below the hit fraction = false alarm fraction line, due to the nature of the hidden state: because the hidden state is more often 0 than 1 ($P_1 < 0.5$), a random spike will have a higher chance of occurring during a period where the hidden state equals 0. Therefore, the false alarm fraction will be higher than the hit fraction for Poisson spike trains.

**Fitting of exponential functions.**   In the results, we fit saturating functions to how *FI* (Eq (8)) depends on different input variables $x \in \{I_{\text{scale}}, r, r_n\}$:

$$FI(x) = FI_{\text{max}}\left(\frac{2}{1 + e^{-\lambda_x (x - x_{\text{offset}})}} - 1\right), \qquad (12)$$

where $r$ is the firing rate of an output spike train, and $r_n = r \cdot \tau_{\text{input}}$ is the unitless firing rate normalized by the switching speed of the hidden state $\tau_{\text{input}}$. We fit parameters $FI_{\text{max}}$ and $\lambda$, and in the case of $x = I_{\text{scale}}$ also $I_{\text{scale,offset}}$ (in the case of $x \in \{r, r_n\}$ the offset value is set equal to 0). To fit these curves, we use Matlab's 'fit' function, which automatically calculates 95% confidence intervals. When the data does not only saturate, but decreases again after the maximum, we include only data up to the maximum.

**Calculation of membrane capacitance and conductance using dynamic IV curves.**   We used the derivation of Badel et al. (2008) [31] to calculate the membrane capacitance ($C_m$) and conductance ($g_m$) for each analysis window. In short, we calculated $\frac{dV_m}{dt}$ from the recorded traces, and calculated the variance of $\frac{I_{\text{inj}}}{C_m^*} - \frac{dV_m}{dt}$ between the values of $-76 \leq V_m \leq -74$ mV for different values of $C_m^*$. The membrane capacitance $C_m$ was determined as the value of $C_m^*$ for which the variance was minimized:

$$C_m = \arg \max_{C_m^*} \text{Var}\left(\frac{I_{\text{inj}}}{C_m^*} - \frac{dV_m}{dt}\right)$$

Next, we defined the membrane current as

$$I_m = I_{\text{inj}} - C_m \frac{dV_m}{dt}$$

and binned the $I_m - V_m$ curve in bins of 5 mV and calculated the average for each bin. A linear fit was made for subthreshold voltage values ($-200 \leq V_m \leq -60$ mV), and the slope was defined as the membrane conductance $g_m$. We excluded files where this fit did not succeed (the value of $g_m$ was found to be negative).

**Spike-triggered average.** The whitened and regularized spike-triggered average (STA) was calculated as

$$STA(t) = (X^T X + \lambda \mu_{X^T X} I)(X^T \rho) \tag{13}$$

where $X$ is a stimulus-lag matrix, where each row is the stimulus vector with a different lag (see [32–35]), $X^T X$ is the correlation matrix and $I$ is the identity matrix. Operator '\' denotes multiplication with the inverse, and $T$ denotes a transpose. Parameter $\lambda$ is a regularization (i.e. smoothing) parameter which was set to 10 and $\mu_{(}X^T X)$ is the mean of the diagonal of the correlation matrix. Finally, $\rho$ denotes the spike train. The resulting STA was normalized with the $L_2$ norm.

Following the derivation of Slee et al. (2005) [36], the inner product of all spike-triggering stimuli with the STA was calculated for each trial (P(stimulus—spike), the posterior distribution), as well as the inner product of the same number of random-triggered stimuli ($P$(stimulus), the prior distribution). With the random-triggered stimuli, the prior distribution of the input was calculated, and compared to the distribution of spike-triggering stimuli (the posterior distribution). With Bayes' law, the shape of the threshold function could be calculated:

$$P(\text{spike}|\text{stimulus}) \sim P(\text{stimulus}|\text{spike})/P(\text{stimulus}) \tag{14}$$

However, if the distributions are not smooth due to limited sampling, the threshold function cannot be calculated. Therefore, the difference in mean between the prior and posterior was calculated for each neuron, and the distribution of means over all neurons was shown.

To assess the variability between the STAs calculated for each cell, we calculated the inner product between all pairs of STAs of inhibitory cells and between all pairs of STAs of excitatory cells. Because excitatory cells fire less, the STAs are based on a lower number of spikes for excitatory cells than for inhibitory cells. This in itself could introduce a higher variability of the STAs. To control for this, we also calculated the STAs for the inhibitory cells based on a comparable number of spikes as for the excitatory cells: we matched each inhibitory trial to an excitatory trial and reduced the number of spikes by only including the first spikes until they had the same amount of spike, and discarding the rest. Subsequently, we calculated the STA based on this reduced number of spikes, normalized them, calculated the posterior and prior for these, and calculated the inner product between all pairs of these STAs.

## Simulations

We performed two types of simulations: an optimal observer for this experiment, the optimal 'Bayesian neuron' [21], and a more biologically realistic exponential integrate-and-fire (expIF) neuron with subthreshold and/or threshold adaptation [22–24].

**Optimal observer: Bayesian Neuron.** Next to the possibility of information estimation in short time windows, the in vitro information transfer method [15] has another advantage: the availability of an optimal observer model. This 'Bayesian neuron' [21] is a spiking neuron model that optimally integrates evidence about the hidden state from the SPNN described

above. It is optimal given an efficient coding or redundancy reduction assumption: it only generates new spikes if those spikes transfer new information about the hidden state, that cannot be inferred from the past spikes in its spike train. In practice, the neuron performs a leaky integration of the input, in order to calculate the log-odds ratio L for the hidden state being 1 (NB Note the similarity with Eq 3):

$$\frac{dL}{dt} = r_{\text{on}}(1 + e^{-L}) - r_{\text{off}}(1 + e^{L}) + I(t) - \theta,$$ (15)

where $r_{\text{on}}$ and $r_{\text{off}}$ are the switching speeds of the hidden state, and $\theta = \sum_{(i=1)}^{N} q_{\text{on}}^i - q_{\text{off}}^i$ is the constant offset of the input generated by the SPNN as before, which is chosen to be equal to 0 in this paper. The neuron compares this log-odds ratio from the input, L, with the log-odds ratio of the hidden state being 1 inferred from its own spike train, G:

$$\frac{dG}{dt} = r_{\text{on}}(1 + e^{-G}) - r_{\text{off}}(1 + e^{G}).$$ (16)

Each time the log-odds ratio based on the input (L) exceeds the log-odds ratio based on the output spike train (G) by an amount $\frac{\eta}{2}$, a spike is fired:

$$\text{if } L > G + \frac{\eta}{2} : \begin{cases} \text{a spike is fired} \\ G \to G + \eta \end{cases}.$$ (17)

For the optimal observer, the parameters of the Bayesian neuron ($r_{\text{on}}$, $r_{\text{off}}$, $\theta$) are the same as the ones chosen for the hidden state and the SPNN for generating the input. As the model is made as an optimal observer for this input, the input does not have to be scaled ($I_{\text{baseline}} = 0$; $I_{\text{scale}} = 1$), making $\eta$ the only free parameter of the Bayesian neuron model. This precision parameter $\eta$ describes the distance between the threshold and the reset of the Bayesian neuron, or in other words, the precision with which G tracks L. This parameter is varied in order to obtain the different firing rates in the Results section (from 0.25 to 6 in steps of 0.25). Note that this neuron model has a form of threshold adaptation: if it did not spike for a long time, G decays to its prior value $G_{\text{prior}} = \log \frac{r_{\text{on}}}{r_{\text{off}}}$. With each spike, G is increased by $\eta$, and more input (larger L) is needed to fire a spike, thereby reducing its firing rate.

The simulated neurons received the same frozen noise input as used in the experiments (see Input current generation), but unscaled ($I_{\text{baseline}} = 0$; $I_{\text{scale}} = 1$). The simulations were performed in Matlab, using a standard forward Euler with a time step of 0.05 ms.

**ExpIF neuron with (sub)threshold adaptation and non-linear I-V curve.** In order to obtain more biologically interpretable results, and to disentangle the subthreshold and threshold effects of adaptation, we used the expIF neuron with subthreshold [22, 23] and/or threshold [24] adaptation. The equations are given by

$$C_m \frac{dV_m}{dt} = \left( g_L(E_L - V_m) + g_L \Delta_T e^{\frac{(V_m - \theta)}{\Delta_T}} + I(t) - w + I_{\{h,K\}} \right)$$ (18)

$$\tau_w \frac{dw}{dt} = a(V_m - E_L) - w$$ (19)

$$\tau_\theta \frac{d\theta}{dt} = \theta_\infty - \theta,$$ (20)

where $w$ describes the subthreshold adaptation and the threshold $\theta$ decays to a steady state

$$\theta_\infty = p * (V_m - V_i) + V_t + K_a * \log\left(1 + e^{\frac{(V_m - V_i)}{k_i}}\right). \tag{21}$$

A spike is defined when $V_m$ passes a cutoff value $V_{\text{cutoff}}$ and is reset to a reset potential $V_r = V_t + 5\Delta_T$:

$$\text{if } V_m > V_{\text{cutoff}} : \begin{cases} \text{a spike is fired} \\ V_m \to V_r \\ w \to w + b \end{cases} . \tag{22}$$

Moreover, we added the following instantaneous currents to the right-hand side of the membrane potential Eq (18) in order to simulate non-linearities in the I-V curve:

$$I_h(V_m) = g_h k_{\infty,h}(V_m)(V_h - V_m) \tag{23}$$

$$k_{\infty,h}(V_m) = \frac{1}{1 + e^{\frac{V^h_{\text{half}} - V_m}{k_h}}} \tag{24}$$

and

$$I_K = g_K k_{\infty,K}(V_m)(V_K - V_m) \tag{25}$$

$$k_{\infty,K}(V_m) = \frac{1}{1 + e^{\frac{V^K_{\text{half}} - V_m}{k_K}}} \tag{26}$$

To assess the effect of adaptation, we simulated 4 parameter regimes: 1) no adaptation, 2) subthreshold adaptation only, 3) threshold adaptation only, and 4) combined adaptation (both subthreshold and threshold adaptation), with the parameters given in Table 2. To assess the effect of a non-linear I-V curve, we simulated 3 parameter regimes: 1) we added an instantaneous hyperpolarization-activated depolarizing current, similar to an h-type current: 'vary $g_h$', 2) we added an instantaneous depolarization-activated hyperpolarizing current, similar to a subthreshold potassium current: 'vary $g_K$', 3) and we varied the leak-conductance: 'vary $g_L$', with the following parameters given in Table 3. Note that for large values of $I_{\text{scale}}$ the simulations diverge: the membrane potential diverges and no further spikes are fired. These simulations are not included in the analyses.

The simulated neurons received the same frozen noise input as used in the experiments (see Input current generation), but with a different scaling (see Tables 2 and 3). Simulations were performed in Brian 2 [37], using a standard forward Euler with a time step of 0.025 ms.

## Results

The goal of this research was to explore the relationship between intrinsic excitability and information transfer. To that end, we first performed 'classical' step-and-hold current clamp experiments in the mouse barrel cortex. Next, we used the 'frozen noise' protocol [15] to measure information transfer together with adaptive properties in these two cell types. After that, we turn to computational modelling to disentangle how different biophysical mechanisms influence information processing in model cells. Finally, we return to the experimental recordings to verify the results obtained from computational modelling.

**Table 2. Parameters for the adaptive expIF model with (sub)threshold adaptation.**

| regime → parameter ↓ | no adaptation | subthreshold adaptation | threshold adaptation | combined adaptation |
|---|---|---|---|---|
| $C_m$ | 50 pF | 50 pF | 50 pF | 50 pF |
| $E_L = V_r$ | -70 mV | -70 mV | -70 mV | -70 mV |
| $g_L$ | 10 nS | 10 nS | 10 nS | 10 nS |
| $\Delta_T$ | 1 mV | 1 mV | 1 mV | 1 mV |
| $\tau_w$ | n/a | varied | n/a | varied |
| $a$ | 0 nS | 4 nS | 0 nS | 4 nS |
| $b$ | 0 nA | 0.0805 nA | 0 nA | 0.0805 nA |
| $\tau_\theta$ | n/a | n/a | varied | varied |
| $p$ | 0 | 0 | 0 | 0 |
| $V_i$ | -67 mV | -67 mV | -67 mV | -67 mV |
| $V_t$ | -63 mV | -63 mV | -63 mV | -63 mV |
| $K_a$ | 0 mV | 0 mV | 5 mV | 5 mV |
| $k_i$ | 5 mV | 5 mV | 5 mV | 5 mV |
| $g_h$ | 0 nS | 0 nS | 0 nS | 0 nS |
| $g_K$ | 0 nS | 0 nS | 0 nS | 0 nS |
| $I_{baseline}$ | 0 nA | 0 nA | 0 nA | 0 nA |
| $I_{scale}$ | varied | varied | varied | varied |

Parameters for Eqs 18–22.

## Information transfer in inhibitory and excitatory neurons

**Excitatory neurons fire at low rates.** Whole-cell recordings were made from pyramidal cells and interneurons in layer 2/3 (L2/3) of mouse barrel cortical slices [13]. Cells were classified as either 'excitatory' or 'inhibitory' based on their electrophysiological responses to a

**Table 3. Parameters for the expIF model with non-linear I-V curve.**

| regime→ parameter ↓ | vary $g_h$ | vary $g_K$ | vary $g_L$ |
|---|---|---|---|
| $C_m$ | 50 pF | 50 pF | 50 pF |
| $E_L = V_r$ | -70 mV | -70 mV | -70 mV |
| $g_L$ | 10 nS | 10 nS | varied |
| $\Delta_T$ | 1 mV | 1 mV | 1 mV |
| $\tau_w$ | n/a | n/a | n/a |
| $\tau_\theta$ | n/a | n/a | n/a |
| $g_h$ | varied | 0 nS | 0 nS |
| $V_{half}^h$ | -82 mV | n/a | n/a nS |
| $k_h$ | -9 mV | n/a | n/a |
| $V_h$ | -30 mV | n/a | n/a |
| $g_K$ | 0 nS | varied | 0 nS |
| $V_{half}^K$ | -60 mV | n/a | n/a nS |
| $k_K$ | 9 mV | n/a | n/a |
| $V_K$ | -70 mV | n/a | n/a |
| $I_{baseline}$ | 0 nA | 0 nA | 0 nA |
| $I_{scale}$ | varied | varied | varied |

Parameters for Eqs 18–25.

standard current-step protocol (Fig 1, see Materials & methods). In response to depolarizing steps, excitatory neurons show strong spike-frequency adaptation, limiting their maximum firing rate (Fig 1 and S1 Table, see also [13]), whereas inhibitory neurons fire at much higher rates.

To measure the information transfer from input current to output spike train, traditionally long ($\sim 1$ hour) experiments were needed to obtain a single mutual information estimate [16–20]. To estimate the information transfer in a shorter time period we used a recently developed method [15] that uses the output of a simulated Poisson neural network (SPNN) to generate the frozen noise current input used in our ex-vivo experiments (Fig 2A, see Materials & methods). Such a frozen noise input constitutes an optimum between giving naturalistic stimuli (as far as possible in an ex-vivo setup, given that we do not have access to the spatiotemporal input distribution a cell would normally receive), being able to assess information transfer, and being able to assess membrane properties (which are typically only stably accessible in slice experiments). The SPNN responds to a randomly appearing and disappearing preferred stimulus or 'hidden state' $x$ (Markov process). This hidden binary state can either be 'on' (i.e. $x = 1$) or 'off' (i.e. $x = 0$), and switches randomly between these states with time constant $\tau_{\text{input}}$. The neurons in the SPNN respond to this hidden state with Poisson-generated spike trains, of which the firing rate depends on the hidden state (i.e. each neuron $i$ responds with a rate of $q_{\text{on}}^i$ when $x = 1$, and a rate of $q_{\text{off}}^i$ when $x = 0$). The mutual information between the input current and the hidden state depends on three properties of the SPNN: the number of neurons ($N$), the average firing rate of the neurons ($\mu_q$), and the time constant of the hidden state ($\tau_{\text{input}}$). We can now compare the mutual information between the input current and the hidden state with the mutual information between the output spike train and the hidden state. This has the advantage that because the hidden state is low-dimensional (it has only two states), the mutual information can be estimated in a short time-window.

Because of the differences in maximum firing rate between the excitatory and inhibitory cells, it was not possible to use the exact same frozen noise input current for the two cell types: $\tau_{\text{input}}$ had to be large for the excitatory neurons (neurons firing at a low rate cannot transfer information about a fast-switching stimulus, so the hidden state had to switch slowly), but this is not the case for the inhibitory neurons (which fire at high rates, so the hidden state can switch fast, i.e. a small value for $\tau_{\text{input}}$ should be used). However, the information in the input could be kept constant by adapting the firing rates of the neurons in the SPNN $\mu_q$ (Fig 2, see also Materials & methods). This resulted in the parameters in Table 1 for the frozen noise experiments to generate the input currents shown in Fig 2.

**Inhibitory neurons show broadband information transfer; Excitatory neurons transfer less information and at low frequencies.** By using the 'frozen noise protocol' as described before (see Materials & methods and [15]), the information transfer from the hidden state to the output spike train of a single neuron can be estimated in a short time window. In order to obtain the information transfer from the input current to the output spike train, we define the unitless fraction of transferred information (*FI*) as the mutual information between the spike train and the hidden state ($MI_{\text{spike train}}$ divided by the mutual information between the input current and the hidden state ($MI_{\text{input}}$, see Eq (8)). The FI quantifies how much information about the hidden state is transferred from the input current to the output spike train, and thus quantifies which fraction of the information is kept during the spike-generating process. In Fig 3, we show the *FI* as a function of the firing rate $r$, for inhibitory (blue) and excitatory (red) neurons, and compare it to the *FI* obtained from the 'Bayesian Neuron' (BN) model [21] for which parameters (see Materials & methods) were optimized for the input generated for the excitatory neurons (pink) or inhibitory neurons (turquoise). Excitatory neurons transfer more

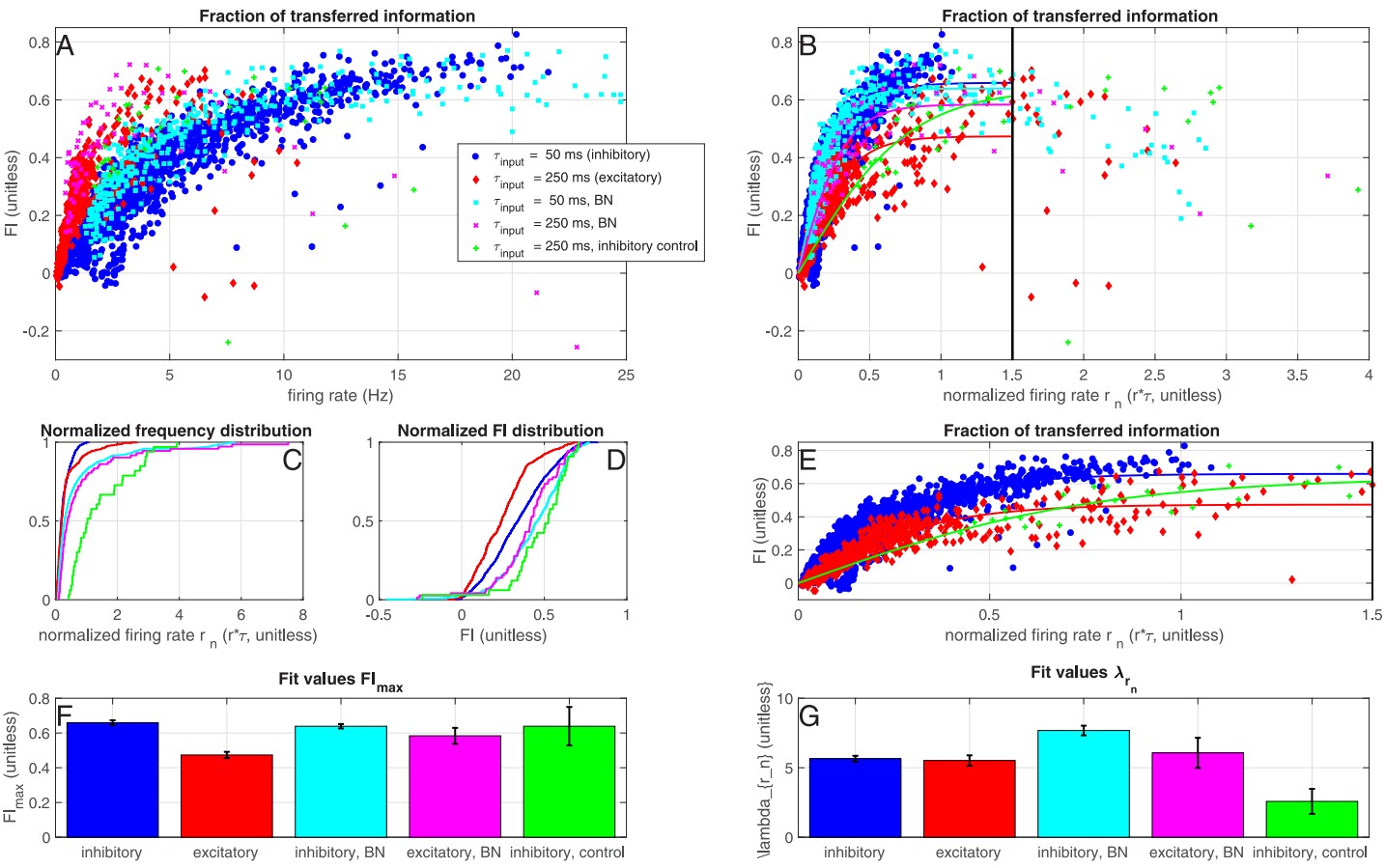

**Fig 3. Inhibitory neurons transfer more information.** A: Fraction of information kept during the spike generating process (*FI*, see Eq (8) as a function of the firing rate, for inhibitory neurons (blue) and excitatory neurons (red). In green, the control experiments where the inhibitory neurons received the input current that was normally given to the excitatory neurons ($\tau_{input}$ = 250 ms). In turquoise and pink, the simulations with the Bayesian Neuron (Materials & Methods, see Table 1 for parameter values). B: Fraction of information kept during the spike generating process, as a function of the normalized firing rate (normalized by the switching speed of the hidden state: $r_n = r \cdot \tau_{input}$, see Table 1). The solid lines denote fits of the data up to a normalized firing frequency of $r_n = 1.5$ (Eq (7), Materials and methods). Colors/markers the same as in A. C: and D: Normalized firing frequency and FI distribution of the spike trains in all conditions. E: Zoom of B. F: and G: Fit values and their 95% confidence intervals (error bars) for parameters $FI_{max}$ (F) and $\lambda_{r_n}$ (G), see Eq (12). Data from 144 excitatory neurons (220 trials), 72 inhibitory neurons (78 trials) and 9 control inhibitory neurons (11 trials). NB Note that even though theoretically $MI \geq 0$, due to our approximation, our estimate of MI can take small negative values (see Materials & methods).

information at low firing rates ($< \sim 8$ Hz) compared to inhibitory neurons. This is due to our choice of slower switching speed (i.e. large $\tau_{input}$) of the hidden state for excitatory neurons: a fast-switching hidden state cannot be properly tracked by neurons firing at a low firing rate (see also [15]). To compare inhibitory and excitatory neurons, we normalized the firing rate of each neuron relative to the switching speed of the hidden state: $r_n = r \cdot \tau_{input}$ (unitless). The *FI* was plotted as a function of this normalized firing rate in Fig 3B. The *FI* increases up to a maximal value at about $r_n = 1.5$, after which the *FI* appears to decrease again. Apparently, at very high firing rates, the transferred information goes down due to too many spikes during $x = 0$. We fitted a saturating function (see Materials & methods) to the measured values, where $FI_{max}$ is the saturation value and $\lambda_{r_n}$ is the rate with which this saturation value is reached (both unitless). We fitted the data up to $r_n = 1.5$, because we do not have a mathematical description for the type of curve that saturates and then dips again (but note that all panels and figures contain all data points, including those for $r_n > 1.5$). In Fig 3E and 3F, the fit values and their 95%

confidence intervals are shown. Inhibitory experimental and BN values saturate around similar values ($FI_{max}$ = 0.65 (0.64–0.66) and $FI_{max}$ = 0.64 (0.63–0.65) respectively), with experiments having a slightly lower rate ($\lambda_{r_n}$ = 5.8 (5.6–6.0) and 7.7 (7.3–8.0) respectively). Excitatory neurons saturate at lower experimental values ($FI_{max}$ = 0.51 (0.48–0.54)) and slightly lower BN values ($FI_{max}$ = 0.58 (0.54–0.63), and the saturation rates are also lower ($\lambda_{r_n}$ = 4.5 (4.0–4.9) and $\lambda_{r_n}$ = 6.1 (5.0–7.2) respectively). This shows that in the case of the excitatory neurons, the experimentally recorded spike trains transmit less information than the spike trains of the BN, whereas in the inhibitory case, the model and experimental spike trains perform similarly. This means that inhibitory neurons perform close to optimal for representing the hidden state, whereas excitatory neurons do not. As a control, we presented the input for the excitatory neurons also to inhibitory neurons (Fig 3, green, $FI_{max}$ = 0.63 (0.52–0.75), $\lambda_{r_n}$ = 2.6 (1.6–3.7)); these inhibitory neurons fired at a higher normalized rate (Fig 3C) and performed better than the excitatory neurons. In conclusion, putative interneurons transfer more information than putative excitatory neurons.

**Inhibitory neurons perform well as classifiers.** The setup with the hidden state makes it possible to show 'receiver-operator curves' (ROCs): we define a 'hit' as a period during which the hidden state was 1 (up-state), in which at least 1 action potential was fired, and a 'miss' as an up-state in which no action potentials were fired. Similarly, we define a 'false alarm' as a period during which the hidden state was 0 (down-state), in which at least 1 action potential was fired, and a 'correct reject' as a down state in which no action potentials were fired. We then define the 'hit fraction' as the number of hits divided by the total number of up-states, and similarly the false alarm fraction for the number of false alarms divided by the total number of down-states. In Fig 4A the results are shown, for the same five conditions as discussed above. For each experiment, a control experiment was simulated by generating a Poisson spike train with the same number of spikes as the original experiment. Note that this 'control' is below the line hit fraction = false alarm fraction because the hidden state is more often 0 than 1 ($P_1 = \frac{1}{3}$). Since the hidden state is longer in the '0' state, the probability that a random spike occurs when the hidden state equals 0 is higher, hence the probability of a false alarm is higher than the probability of a hit.

Inhibitory neurons perform comparably to the BN, as shown in Fig 4, whereas the excitatory neurons perform less optimally than their model counterparts. We performed control experiments where input currents generated for excitatory neurons were injected into inhibitory neurons, (green triangles in Figs 3 and 4). The results suggest that interneurons perform comparably to (on the same curve as) excitatory neurons, but with a lower discrimination threshold (i.e. with a higher firing rate), which is in agreement with our previous observation that inhibitory neurons responded with a higher firing rate than excitatory neurons. Note that inhibitory neurons fire slightly less spikes during the up-states (Fig 4B) and the normalized firing rate in the up-state is somewhat lower for the inhibitory neurons (Fig 4F). Since the excitatory neurons fire more spikes during the down states (Fig 4C and 4G), this corresponds to a lower efficiency for excitatory neurons and a worse performance on the binary classification task (Fig 4A). Indeed, the number of spikes per down state (Fig 4C) and normalized firing rate in the down state (Fig 4G) differs between inhibitory and excitatory neurons (S2 and S3 Tables). Note that most 'incorrect' spikes are actually fired shortly after a down switch (Fig 4H–4K), so they might be 'correct' spikes that were a few milliseconds too late. In conclusion, putative interneurons are better binary classifiers than putative excitatory neurons.

**Dynamic threshold of both neuron types.** To assess how intrinsic properties of the putative interneurons and pyramidal cells correlate with their information transfer capabilities in this setup, we next assess the threshold adaptation of these neuron types. In Fig 5, we show the

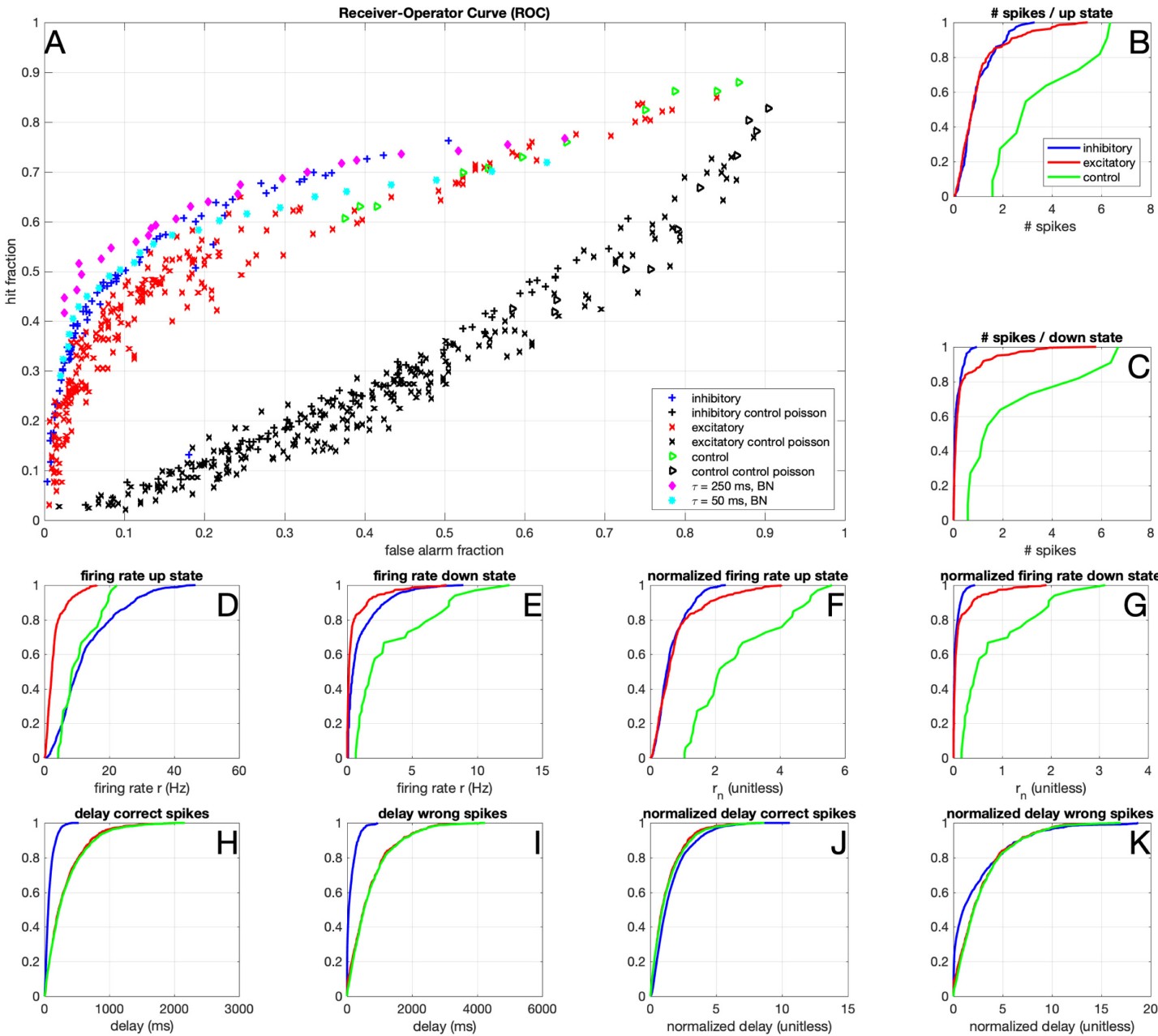

**Fig 4. Binary classification.** A) Receiver Operator Curve (ROC), where the hit rate was defined as the fraction of up-states, in which at least 1 action potential was fired. Similarly, the false alarm rate was defined as the fraction of down-states, in which at least 1 action potential was fired. In black the results for Poisson spike trains with firing rates matched to those of the experimental/simulation conditions are shown. B) Distribution of the number of spikes per period where the hidden state was 1 (up state), for inhibitory neurons (blue) and excitatory neurons (red). C) Same as B), but for periods where the hidden state was 0 (down state). D) Firing rate $r$ distribution in the up-state. E) Firing rate $r$ distribution in the down-state. F) Normalized firing rate $r_n$ distribution in the up-state G) Normalized firing rate $r_n$ distribution in the down-state. H) Delay (in ms) of each correct spike since the state switches from down to up. I) Delay (in ms) of each incorrect spike since the state switches from up to down. J) Normalized delay (delay/$\tau$, unitless) of each correct spike since the state switches from down to up. K) Normalized delay of each incorrect spike since the state switch from up to down. The results of hypotheses test for A-F are in S2 and S3 Tables. Data from 144 excitatory neurons (220 trials), 72 inhibitory neurons (78 trials), and 9 control inhibitory neurons (11 trials).

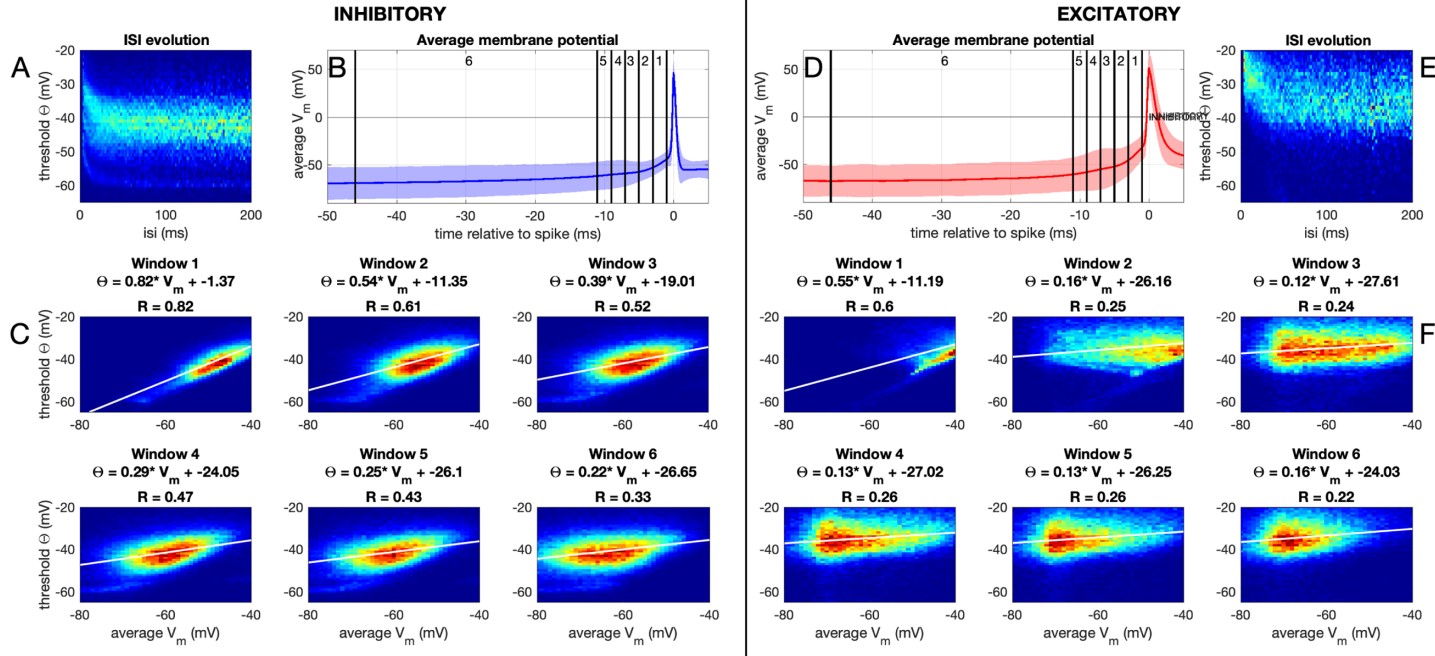

**Fig 5. Dynamic threshold.** A-C) Inhibitory neurons. D-F) Excitatory Neurons. A) Distribution of membrane potential threshold values (see Materials & methods) for each inter-spike interval (ISI); normalized per ISI. B) Average spike shape (shaded region denotes standard deviation). Vertical lines denote the windows in C. C) Heatmap and regression for the relation between the threshold and the average membrane potential in the given window. D-F) Same as in A-C, but for excitatory neurons. This is all in the Frozen Noise protocol, for threshold behaviour in the current-clamp step-and-hold protocol, see S1 Fig.

threshold behaviour of the inhibitory and excitatory neurons. The membrane potential threshold of each spike was determined based on the method of [24] (see Materials & methods). We show the distribution of the membrane potential as a function of the inter-spike interval (ISI, Fig 5A and 5E). For both inhibitory and excitatory neurons, the membrane potential threshold goes up with short ISIs, as expected, and for long ISIs the threshold is low. This effect has a long time scale (at least several tens of milliseconds), longer than expected based on the relative refractory period alone (typically less than ten milliseconds). The thresholds of excitatory neurons are almost 10 mV higher than those of inhibitory neurons (Fig 5A and 5E and S1A–S1C Fig). Next to the ISI, the threshold also depends on the history of the membrane potential (Fig 5C and 5F): we calculated the regression between the action potential threshold and the average membrane potential in different windows preceding the spike. There is a strong correlation between the threshold and the membrane potential immediately preceding the spike for both neuron types, which reduces gradually with time before the spike. However, for both neuron types, some relation between the membrane potential several tens of milliseconds before the spike and the threshold is still visible. The current clamp step protocol (S1 Fig) confirms the overall higher thresholds (S1A and S1B Fig) and strong spike-frequency adaptation (S1D Fig) for excitatory neurons. The threshold adaptation rate however, shows significant differences between fast spiking and regular spiking neurons at current injection intensities ranging from +240 to +320pA, while they do not show significant changes at lower or higher intensities, possibly due to low firing rates or reaching a steady state firing rate. (S1C Fig, S1 Table).

So in conclusion, both inhibitory and excitatory neurons show a dynamic threshold behaviour, with inhibitory neurons having much lower thresholds, so they can fire at high rates,

whereas the dynamic threshold of excitatory neurons promotes low-frequency firing and shows stronger adaptation.

## Information transfer in simulated neuron models

In the experimental data, we saw that both fast-spiking interneurons and regular spiking excitatory neurons transfer a significant amount of information about the hidden state, not much less than the optimal Bayesian neuron, as they adapt their spike threshold to the dynamics of the stimulus. The goal of this research was to explore the relationship between intrinsic excitability and information transfer. The Bayesian neuron that is optimal for this task has two properties that distinguish it from a standard integrate-and-fire model: 1) spike-frequency adaptation and 2) a non-linear I-V curve. To untangle how these mechanisms influence information transfer, we turn to computational modelling. We use an exponential integrate-and-fire (expIF) model and adapt both its adaptation and the shape of the IV-curve, to explore how these affect information transfer.

**Information transfer in neuron models with (sub)threshold adaptation.** In the previous section, we saw that both inhibitory and excitatory neurons show a dynamic threshold behaviour, suggesting that both cell types have in theory the adaptation mechanisms that can influence information transfer, as is also present in the Bayesian Neuron. In biophysical models, spike-frequency adaptation can be implemented in different ways [11]. Particularly, in the expIF model, it has been implemented as either a subthreshold process [22, 23] or as an adaptation of the spike threshold [24]. We research the effects of these two types of adaptation on the information transfer in the aforementioned mutual information protocol.

In Fig 6C, we first note that the 'slow' input to the excitatory neurons is apparently more difficult to transfer than the 'fast' one: the exact same expIF model transfers less of the 'slow' input information (red) than of the 'fast' one (blue). Next, in Fig 6D–6G, we show that adding threshold adaptation does not increase the amount of information that is transferred by the neuron. However, it does shift its working range towards higher values of the input amplitude $I_{\text{scale}}$, effectively increasing its working range. Contrasting, in Fig 6H–6K, we show that adding subthreshold adaptation does increase the maximum information transfer when it is properly tuned, i.e. when the time constant of adaptation fits the input properties. However, too slow adaptation suppresses the firing rate too much (Fig 6J and 6K), resulting in a reduction of information transfer.

We ask whether the effects on information transfer are a result of a higher firing rate, or of a better detection. Therefore, we turn to the ROC curves discussed before. In Fig 7 we show that both forms of adaptation do not change the shape of the ROC curve. However, we do note that for the 'slow' input, the expIF neuron performs much worse than both the Bayesian neuron and the experimentally recorded neurons.

In conclusion, we see that subthreshold, but not threshold, adaptation can increase the maximum information transfer. Threshold adaptation, on the other hand, can increase the working range of the neuron. Moreover, an expIF neuron performs worse than both the Bayesian neuron and the experimentally recorded neurons. Since the Bayesian neuron differs from the expIF model in its IV curve, we next determine how information transfer is influenced by the shape of the IV curve.

**The shape of the IV curve.** We assess the effects of changing the shape of the I-V curve (the right-hand side of the membrane voltage equation). The Bayesian neuron, tailor-made to transfer information efficiently for this type of input, has two features that distinguish it from a classical integrate-and-fire model: an adaptation mechanism, discussed in the previous paragraph, and a non-linear IV-curve, as can be seen in Fig 8A: the amplitude of the IV-curve

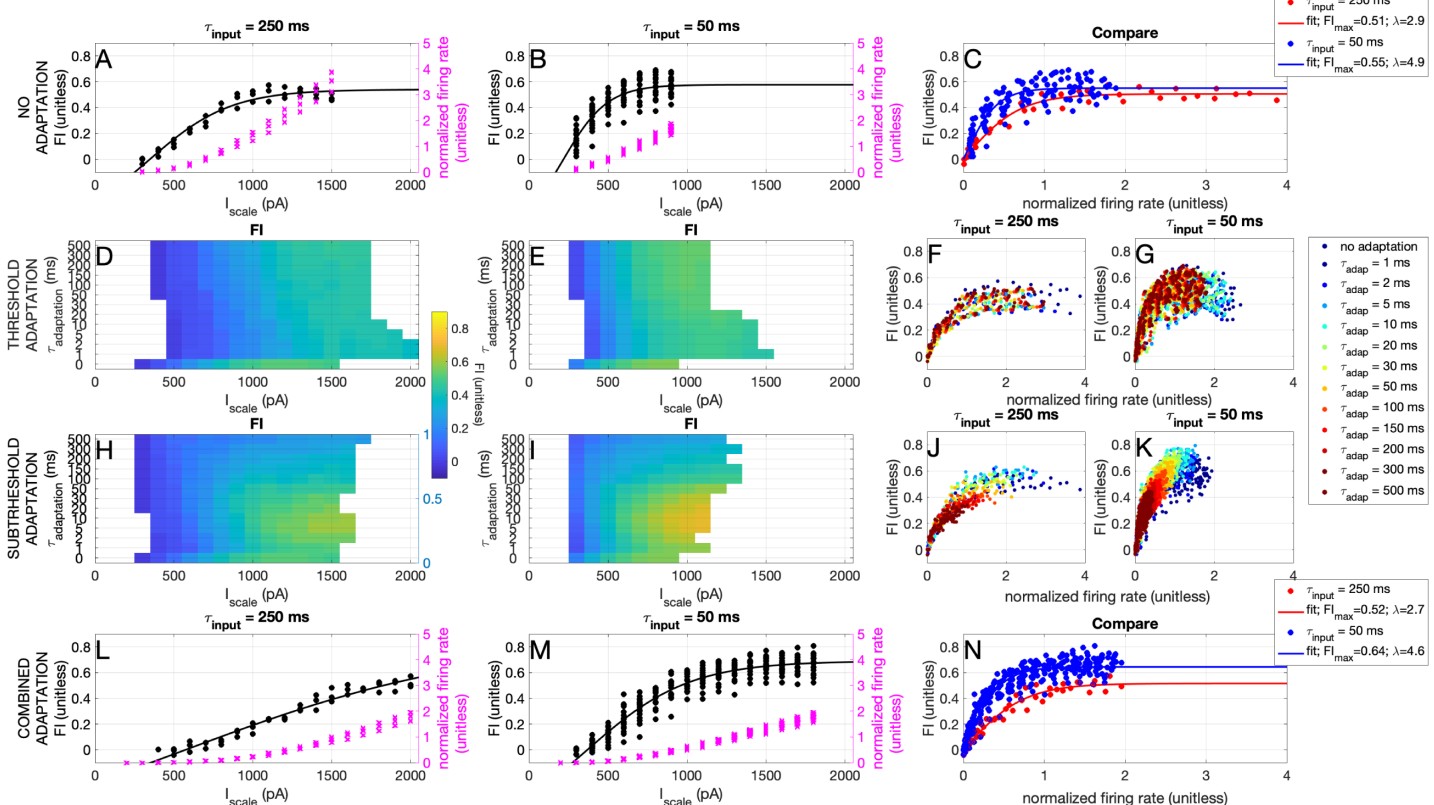

**Fig 6. Effects of the dynamics of adaptation on information transfer.** A-C) No adaptation. D-G) Threshold adaptation. H-K) Subthreshold adaptation. L-N) Combined adaptation. Left column: 'slow' input (time constant hidden state $\tau_{input}$ = 250 ms). Middle column: 'fast' input (time constant hidden state $\tau_{input}$ = 50 ms). Right column: 'slow' and 'fast' input together. The time constant of the adaptation, $\tau_t extrmadap$, corresponds to $\tau_w$ for subthreshold adaptation or $\tau_\theta$ for threshold adaptation (Eq (18)) A) Fraction of information (*FI*, black) and normalized firing rate $r_n$ (pink) as a function of the input amplitude $I_{scale}$ for the expIF model without adaptation. B) same as A but for the 'fast' input. C) Fraction of information as a function of the normalized firing rate for the 'slow' (red) and 'fast' (blue) input for the expIF model without adaptation. D) Fraction of information (colorbar) as a function of the input amplitude $I_{scale}$ for the expIF model with threshold adaptation with different adaptation time constants $\tau_{adap}$ (vertical axis) receiving the 'slow' input. E) Same as D, but for the 'fast' input. F) Fraction of information as a function of the normalized firing rate for the 'slow' input or the expIF model with threshold adaptation with different adaptation time constants $\tau_{adap}$ (colours). G) Same as F, but for the 'fast' input. H-K) Same as D-G, but for the expIF model with subthreshold adaptation. L-N) Same as A-C, but for the model with both threshold ($\tau_{adap}$ = 1 ms) and subthreshold ($\tau_{adap}$ = 10 ms) adaptation.

increases exponentially when moving away from the steady-state value (dotted vertical line). We add such non-linearities in the expIF neuron in a biologically realistic way, to see how they would influence the classification (ROC curve) and the information transfer. Firstly we model the effects of the suppression of hyperpolarization (i.e. increasing slope of the IV curve when hyperpolarizing the cell) by adding an instantaneous 'h-current' to the expIF neuron (see Materials & methods), as shown in Fig 8B. Next, we model the effects of the suppression of depolarization (i.e. increasing slope of the IV curve when depolarizing the cell) by adding an instantaneous subthreshold potassium current, as shown in Fig 8C. Finally, we also change the overall slope, but not the shape of the IV curve, by changing the leak conductance of the neuron, as shown in Fig 8D. In Fig 8F and 8J, we show that adding the 'h-current' (with conductance $g_h$) does not change the shape of the ROC curve. However, its effect is similar to lowering the detection threshold (i.e. the values shift over the curve towards higher hit and false alarm fractions). On the contrary, the addition of the potassium current (with conductance $g_K$) does not change the shape of the ROC curve, but its effect is similar to an increase in

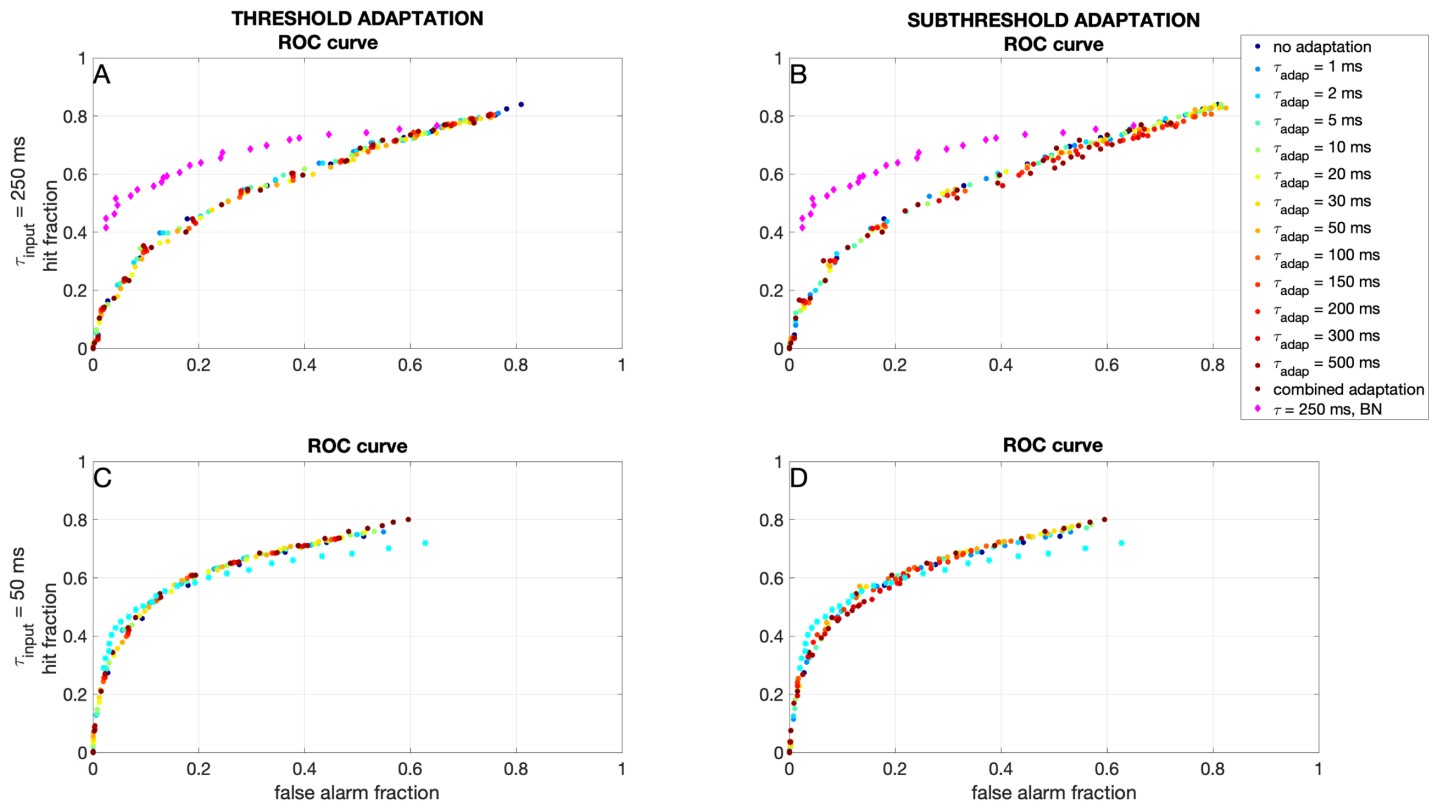

**Fig 7. Effects of the dynamics of adaptation on binary classification.** A) Receiver Operator Curve (ROC) (see also Fig 4) for the expIF neuron with threshold adaptation (colors denote time constant) receiving 'slow' input. Note that the adaptation does not change the shape of the ROC curve, and that the neuron performs much worse than the Bayesian neuron (pink). B) Same as A, but for subthreshold adaptation. C) Same as A, but for the neuron receiving 'fast' input and the response of the Bayesian neuron in turquoise. D) Same as C, but for the neuron with subthreshold adaptation.

the detection threshold (i.e. the values shift over the curve towards lower hit and false alarm fractions, Fig 8G and 8K). Changing the overall slope of the IV curve (i.e. the 'leak conductance' $g_L$) does change the shape of the ROC curve (Fig 8H and 8L): for the slow input current ($\tau_{\text{input}} = 250$ ms) it needs to be tuned to a lower value ($g_L \approx 1$ nS) than for the faster input current ($\tau_{\text{input}} = 50$ ms) for optimal information transfer and classification.

The effects seen in the ROC curves are confirmed by the information transfer measurements: in Fig 9A, 9D, 9G and 9J we show that adding an 'h-current' can strongly increase the information transfer of the expIF neuron, by increasing its firing rate. Adding a subthreshold instantaneous potassium current shows the opposite effect: it decreases both firing rates and information transfer (Fig 9B, 9E, 9H and 9K). Finally, the slope of the IV curve needs to be matched to the input statistics: the slow input needs a flatter IV-curve (lower $g_L$) than the fast input for information transfer (Fig 9C, 9F, 9I and 9L).

In conclusion, we saw that the recorded excitatory neurons perform better for slow input than the expIF model with or without adaptation; in fact, these neurons perform similarly to the optimal simulated model (the Bayesian Neuron). Recorded inhibitory neurons perform close to optimal for fast input, a result well captured with an expIF model that includes an adaptation mechanism. Threshold adaptation increases the working range of the expIF model, but does not increase, or even slightly reduces, the amount of transferred information. Subthreshold adaptation, on the other hand, does not increase the working range but does increase

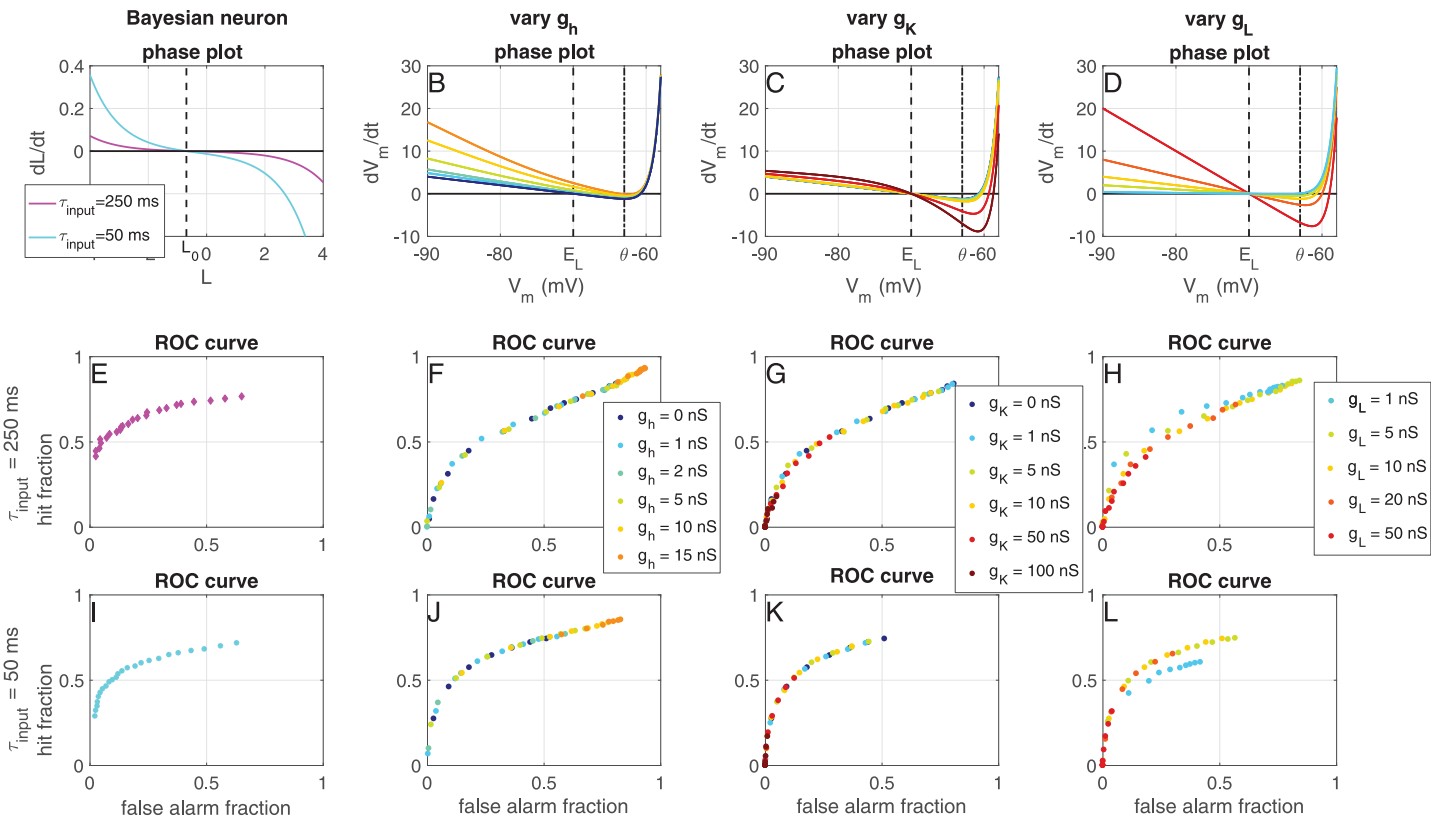

**Fig 8. Effects of the IV-curve shape on binary classification.** A-D) IV-curves (i.e. right-hand side of the membrane voltage equation) for the Bayesian Neuron (A) and for the expIF neuron (without adaptation) with added 'h-current' (B), 'subthreshold potassium current' (C) and while varying the membrane conductance (D), colours denote conductances (see legends below). E-H) Receiver Operator Curve (ROC) (see also Fig 4) for the expIF neuron with different IV curve shapes (colors denote time constant) receiving 'slow' input. Note that only $g_L$ changes the shape of the ROC curve. I-L) Same as E-H, but for the neuron receiving 'fast' input.

the maximum transferred information if correctly tuned. Neither form of adaptation changes the shape of the ROC curve. The slope of the IV curve does play an important role in the information transfer and needs to be tuned to the statistics of the input. To check this conclusion, we will next assess this statement in our experimental recordings.

**Back to the recordings: Dynamic IV curve unravels the relationship between membrane conductance and information transfer.** We assess the relation between the slope of the IV-curve and the information transfer, by determining the dynamic IV-curve [31] for each of our recordings (see Materials and methods). In Fig 10, we show the fraction of transferred information (*FI*) as a function of the membrane conductance $g_m$ and membrane capacitance $C_m$ (Fig 10A and reffigdynIVB) and as a function of the membrane time constant $\tau_m$ (Fig 10C and 10D). As in the expIF model simulations, we can conclude that the fraction of transferred information depends on the slope of the IV-curve: we see a clear inverse relation between membrane conductance and transferred information. However, the recorded neurons show quite a large variability of intrinsic properties, in particular the regular spiking excitatory neurons. To assess how this large heterogeneity of excitatory neurons influences their response properties, we calculate their spike-triggered averages.

**Back to the recordings: Response heterogeneity of the Spike-Triggered Average.** In Fig 11, we show the normalized spike-triggered averages (STAs) for spikes of inhibitory (A and E) and excitatory neurons (C). The filter was whitened and regularized (see Materials &

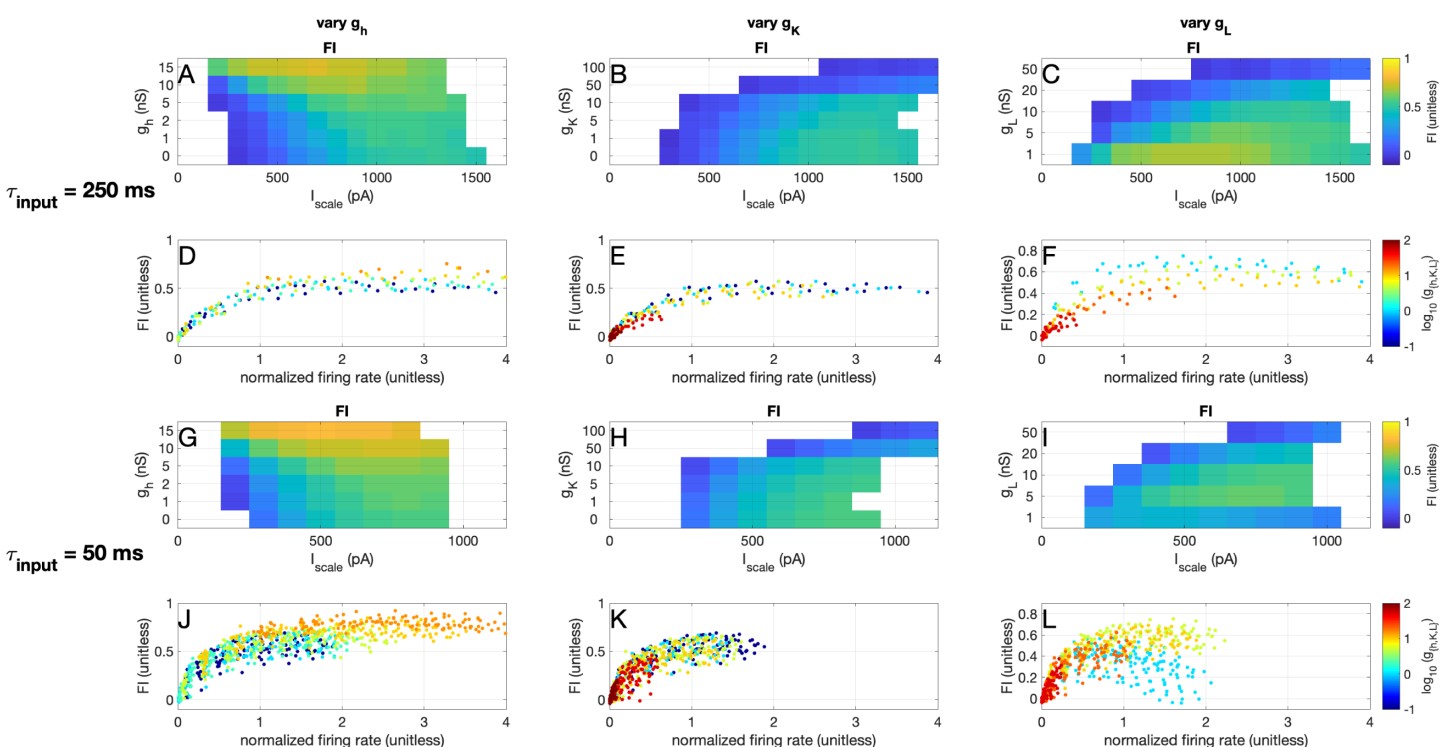

**Fig 9. Effects of the IV-curve shape on information transfer.** A) Fraction of transferred information $FI$ as a function of the input amplitude $I_{scale}$ for the expIF model with added instantaneous 'h-current' with different values of conductance $g_h$ (vertical axis) receiving the 'slow' input. B) Same as A but for the subthreshold instantaneous 'potassium current' ($g_K$). C) Same as A but for the leak conductance $g_L$. D-F) Fraction of information as a function of the normalized firing rate for the 'slow' input or the expIF model with different IV shapes (colours). G-L) Same as A-F but for the 'fast' input.

methods). Note that the filters contain an acausal part, because of the autoroccelations in the input current: the spikes of the SPNN were convolved with an exponential kernel of 5 ms, and this is reflected in the STAs. Next, the projection values of spike-triggering and random currents were calculated (see Fig 11B for an example for 1 cell), and the distance between the means of the distributions for random and spike-triggering currents was calculated for each cell (Fig 11D). The average STAs for all inhibitory (Fig 11A, blue) and excitatory (Fig 11C, red) neurons were quite similar, but the traces for individual neurons (grey lines) were much more variable for excitatory neurons than inhibitory neurons. This indicates that the excitatory neurons have a higher variability in their feature selectivity of incoming current stimuli than inhibitory neurons, as was expected from the higher intrinsic variability discussed in the previous section. However, it is also possible that this is an effect of the lower number of spikes available for excitatory neurons. To control for this possibility, we calculated the STAs for spike trains of inhibitory neurons, where the number of spikes was reduced to match an 186 excitatory trial (Fig 11E, brown). For all three groups (inhibitory, excitatory, and inhibitory control spike trains) we calculated the inner product between all calculated STAs. Fig 11F shows the distributions of these inner products, and it is clear that both inhibitory full and control spike trains are much less variable (inner product closer to 1) than the excitatory spike trains (two-sample Kolmogorov-Smirnov test E-I $p = 0$, E-C $p < 1e − 223$, I-C $p < 1e − 228$). The distribution of all distances between the means is shown in Fig 11D. The distances between the distributions, measured in standard deviations of the prior (random triggered currents) distribution, are much higher for excitatory neurons than for inhibitory neurons,

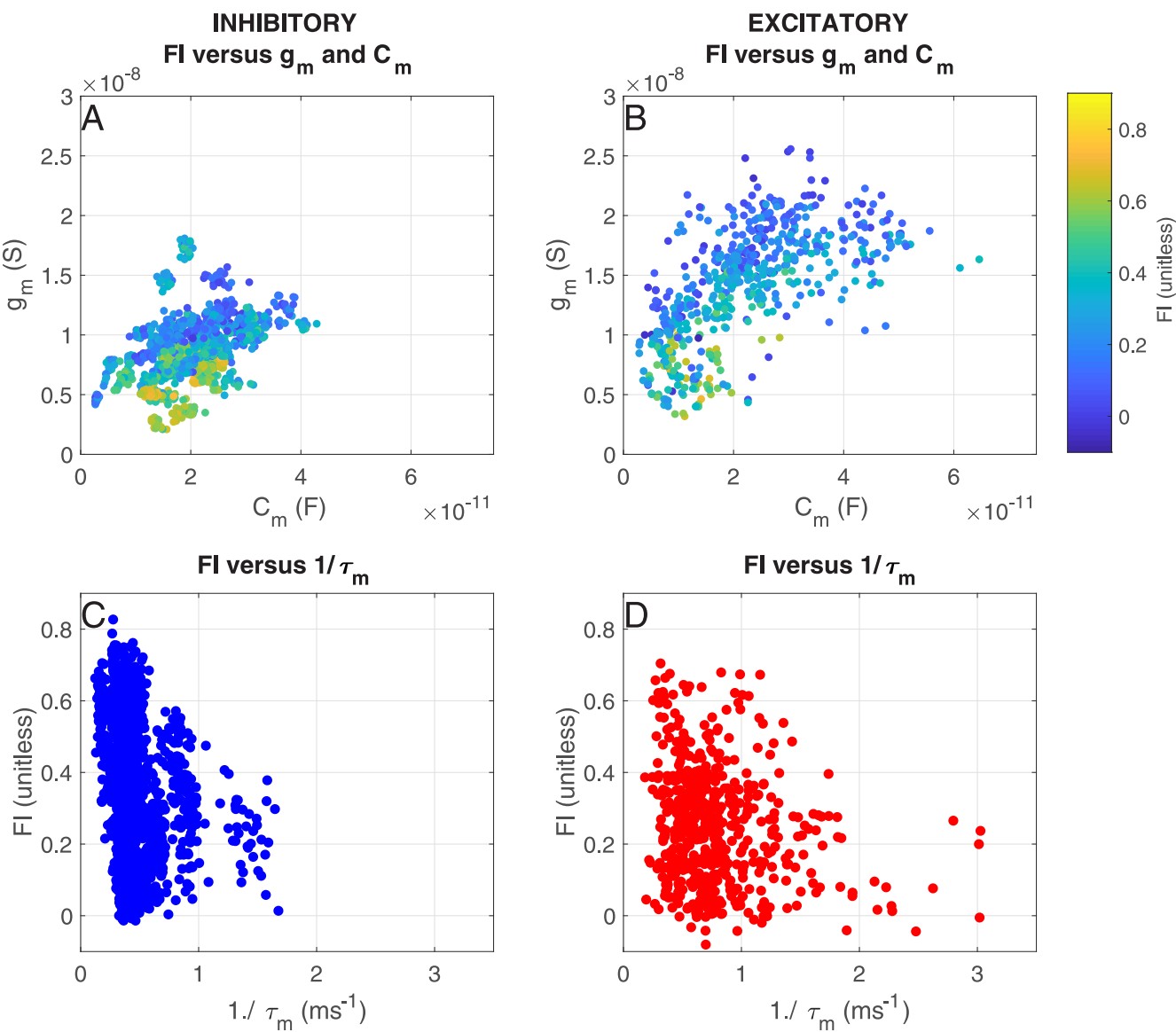

**Fig 10. Effects of the dynamic IV-curve shape on information transfer of recorded neurons.** A) Fraction of transferred information *FI* as a function of the membrane conductance $g_m$ and capacitance $C_m$ for inhibitory neurons. B) Same as A), but for excitatory neurons. C) Fraction of transferred information *FI* as a function of the inverse of the membrane time constant $\tau_m$ for inhibitory neurons. D) Same as C), but for excitatory neurons.

indicating that excitatory neurons are more selective (p-values two-sample t-test: E-I $p < 1e - 28$, E-C $p < 1e - 24$, I-C $p = 0.14$). In conclusion, excitatory cells fire less than inhibitory cells and are therefore more selective, but at the same time, there is more variability between excitatory neurons in what input features they respond to than between inhibitory cells.

## Conclusion and discussion

We measured the differences in information transfer between (putative) inhibitory interneurons and excitatory pyramidal cells in the cerebral cortex. We utilised a technique in which the input current was generated by an Simulated Poisson Neural Network (SPNN), with each

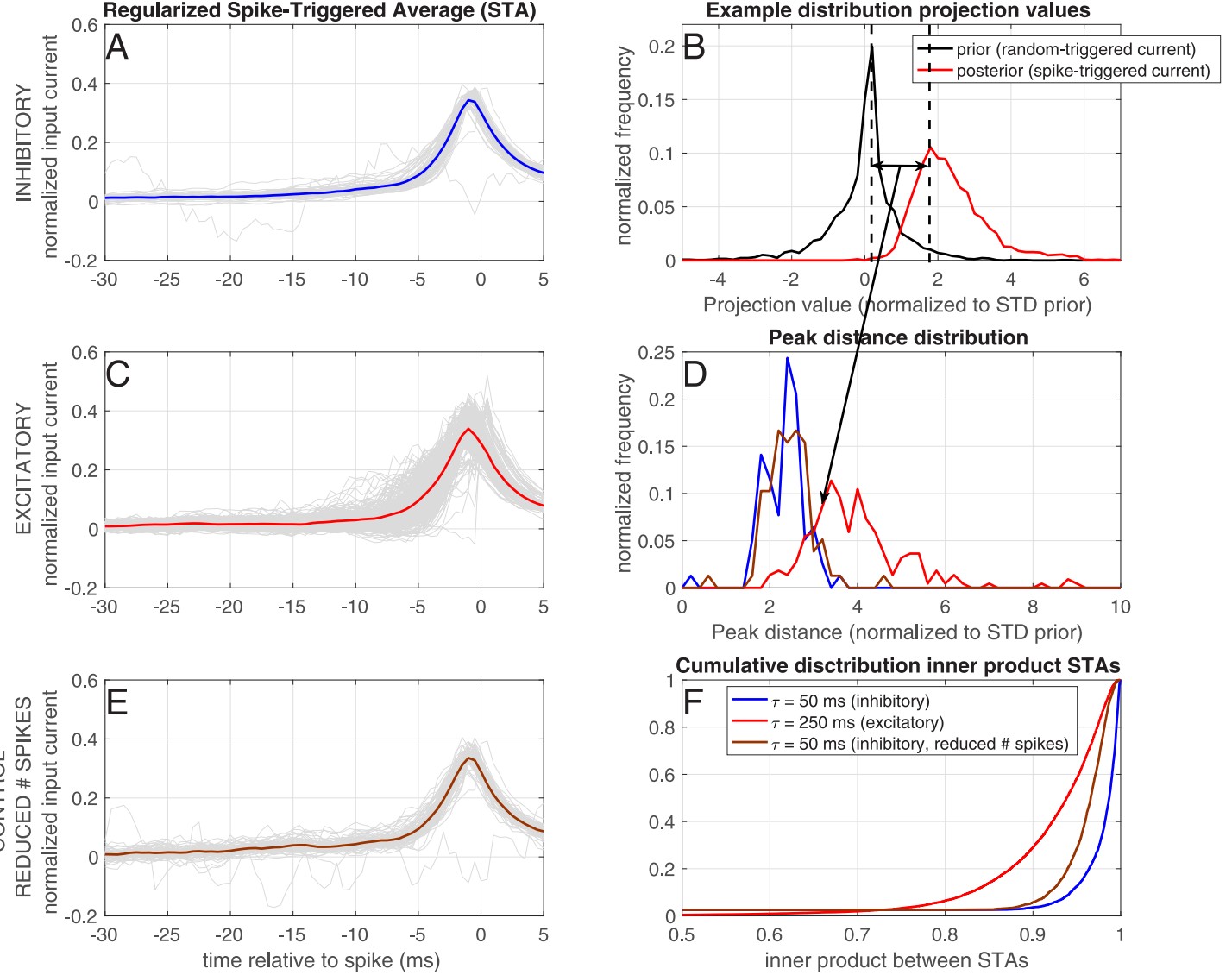

**Fig 11. Linear filtering properties of recorded neurons.** A) Whitened and regularized (see Materials and methods) Spike-Triggered average (STA) for inhibitory neurons. The STAs for individual neurons are shown as thin grey lines, and the average over neurons is shown as a thick coloured line. B) Example of a prior (random triggered, black line) and posterior (spike-triggered, red line) distribution of stimulus projection values for a single inhibitory neuron. C) Same as A), but for excitatory neurons. D) Distribution of the differences between the means (see arrow in B) between the prior and posterior distribution over all neurons.-E) Same as A), but for the reduced (i.e. fewer spikes) spike trains of the inhibitory neurons. F) Distribution of the inner products between the STAs for the three groups (note that because the STAs are normalized by the L2-norm, the maximal value of the inner product is limited to 1). Data from 144 excitatory neurons (220 trials) and 72 inhibitory neurons (78 trials).

artificial cell firing Poisson spike trains whose firing rate was modulated by the absence or presence of the stimulus [15]. We discovered that excitatory cells are more selective: they respond more sparsely and transfer less information. Inhibitory neurons exhibit a near-optimal response, transferring a great deal of input-related information at relatively rapid rates. In a computational model, the mechanisms that can explain such differences in information transfer were investigated. We evaluated the effects of (sub)threshold adaptation and the IV curve's shape. We discovered that adaptation increases information transfer (subthreshold adaptation) and the working range (threshold adaptation). In addition, the shape of the IV-

**Table 4. Conclusions of the Exponential IF model simulations.**

| Mechanism | Max information transfer | Working range | ROC curve |
|---|---|---|---|
| threshold adaptation | unchanged/reduced | increased / shift to higher amplitudes | unchanged |
| subthreshold adaptation | increased (if tuned properly) | unchanged | unchanged |
| steepness IV curve ($g_L$) | increased (if tuned properly) | depends on tuning | better detection if tuned properly |
| hyperpolarized part IV curve ($g_h$) | increased at the cost of higher firing rate | shift to lower amplitudes | unchanged shape shift towards higher rates |
| depolarized part IV curve ($g_K$) | decreased | shift to higher amplitudes | shape unchanged shift towards lower rates |

curve plays a crucial role in determining the information transfer: the slope must correspond to the input characteristics, and the suppression of hyperpolarization, such as by a 'h-current,' can increase the information transfer. The effects are summarised in Table 4. Although the current experimental data does not permit an explicit test of the effects of (sub)threshold adaptation and/or 'h-current,' the relationship between information transfer and the slope of the (dynamic) IV-curve (the membrane conductance) can be evaluated. As predicted, we observe an inverse relationship between membrane conductance and information transfer. Finally, we find that both the intrinsic (membrane conductance) and response (STA, FI) properties measured cells show a large heterogeneity, with excitatory neurons exhibiting more heterogeneity compared to inhibitory neurons.

The method that was used for measuring the information transfer in the ex-vivo experiments [13, 15] is fast and efficient (an estimate of the information transfer can be made within a few minutes), but this comes at a cost. The method uses a reduction in entropy of the input (a binary hidden state) to speed up the information calculation. This excludes for instance the possibility of a graded response (in our setup, a stimulus is either present or not, there is no measure of 'how much' it is present). Also, even though in the decoding step the integration of the log-likelihood $L$ does include the timing of individual spikes, the method does not consider 'pattern codes' or 'phase codes'. The decoder assumes an inhomogeneous Poisson process, so essentially a time-varying rate code: spikes have to be aligned to the the periods when the hidden state is 1, but a relative spike timing code cannot be detected. This means that the setup might perform better for 'type 1' neurons (or 'integrators') relative to 'type 2' neurons (or resonators), 'type 3' neurons (or 'differentiators') [38–40] or bursting cells [41]. This remains a topic for future studies. More generally, finding a representative input is always somewhat problematic in ex-vivo experiments, because the exact spatiotemporal distribution of input to a neuron is often not known. Traditionally, either 'step-and-hold' or white noise inputs are used. In that sense, the Poisson spike trains convolved with an exponential kernel are not less naturalistic than other classical inputs. Also, many other types of analysis (i.e. reverse correlation analysis) are possible with this setup. A dynamic clamp (in which a conductance rather than a current is injected) process would be a more naturalistic extension of the current setup.

Next to the limitations induced by our limited knowledge of the input a neuron receives in vivo, our experiments also suffered from the following limitation: because the inhibitory and excitatory neurons show very different firing rates, we could not give them the exact same input. The low spike rates of in particular our excitatory cells is a well documented property of excitatory neurons in (barrel) cortex [42]. Inhibitory neurons are known to have higher, but still sparse firing patterns [43] (for a review, see [44]). We have aimed to match the switching speed of the hidden state to the 'natural' firing frequencies of inhibitory and excitatory neurons, but this came with the cost that the two neuron types did not receive the exact same input. This makes the comparison of the information transfer between excitatory and inhibitory less trivial. We have opted for a trade-off between switching speed and the number of

spikes in the SPNN (effectively the amplitude of the input, see Fig 2), which like discussed in the previous paragraph comes with the assumption that the neurons are effectively 'integrators'. It might be, that neurons are in fact doing something different alltogether. This is often an inherent problem with measuring information transfer: a decoding algorithm or task needs to be assumed and the choice of decoding algorithm is not trivial.

A third limitation is that we do not consider the role of the neuron in the circuit it is embedded in. It is possible, that the roles of excitatory and inhibitory are very different in cortical circuits and that this necessitates different electrophysiological properties. For instance, one theory suggests that fast inhibition is needed for predictive coding [45–48]. We find that excitatory neurons, the neurons that connect not only locally but also across layers and areas, are more *selective* (i.e. have both a higher feature (Fig 11D) and membrane potential (Fig 5A and 5E) threshold), as well as more *variable* in what stimulus features they represent (Fig 11A, 11C, 11D and 11F). It is possible, that because excitatory neurons represent a more diverse set of features that their spikes are more informative in a population code, as they span a higher-dimensional feature space. This would suggest that the inhibitory neurons 'gate' the message to the next processing layer: the strong compression of information in excitatory neurons and the higher activity levels in inhibitory neurons, would then result in a specific and sparse code. Indeed, the activity of excitatory neurons in barrel cortex in vivo is extremely sparse, and they have very narrow receptive fields [49]. Moreover, it has been shown that adaptive threshold model neurons are more informative for high temporal precision and low noise than fixed threshold model neurons [50]. Therefore, it is likely that especially noisy stimuli with low temporal precision are suppressed.

Finally, we do not consider the specific tasks neurons in the barrel cortex are performing. Rodents explore their surroundings by whisking, i.e. moving the whiskers back and forth with different speeds, depending on the task (about 4–25 Hz [51]). Different neural pathways are thought to be responsible for carrying information about the location of objects (object localization) and the properties of these objects (texture discrimination) [52, 53], but these both end up in the somatosensory or barrel cortex. It has been shown recently that the information from both pathways is encoded very sparsely and temporally precisely [42, 54–61]. How the information from the different pathways is encoded exactly and what type of coding (rate, phase, pattern, local, distributed) is used for what task is still under active investigation, but a tentative conclusion is that in the context of texture discrimination, the task of in particular the cortex (as opposed to earlier processing stages) is the accumulation of evidence over time (for a recent review see [62]). Our paradigm is well suited to measure such integration over time, but of course it remains to be validated what the relevant time scales are and what exactly the input a cortical neuron would receive looks like. In the context of object localization, the whisker response is thought to be not only sparce and temporally precise, but also quite reliable, whith excitatory neurons receiving strong simultaneous inhibitory and excitatory input (for recent reviews see [63, 64]), which would hint at a 'gating' coding scheme at the population level as discussed above. So the coding scheme assumed in our experimental context might be more relevant for texture discrimination than for object localization.

In this combined experimental-modelling approach, we observed that of the mechanisms we varied the slope of the relation between the input current and the membrane potential shows the strongest correlation with the amount of information a neuron can transfer. This slope determines essentially the working range of the neuron: the range of input currents to which a neuron can show a graded response. Therefore, the statement that the slope of the I-V curve needs to be tuned to the statistics of the input, basically means that the working range of the neuron needs to be tuned to the range of the input, a process called gain modulation [3, 7]. For complex, non-linear systems such as neurons, the effective gain can depend on the type of

input that the it receives: fluctuating, noisy input can result in a different gain than the classical 'step-and-hold' inputs [65, 66]. Essentially, the nervous system needs mechanisms to adapt the slope of its input-output relation when the input statistics change, at different timescales. It remains an object for future studies how, and how fast, neurons can do this and what mechanisms (and neuromodulators) the nervous system uses for such adaptation.

It has been shown repeatedly, that the spiking behaviour of cortical neurons can be fitted relatively well with a simple threshold model with an extra feedback variable [22, 24, 67–77]. The heterogeneity in such cell properties has been investigated in excitatory [78] and inhibitory [79] cells. With this manuscript, we add a functional dimension to these basic properties of cortical spike initiation: We show how different mechanistic features of cortical cells can influence their information transfer. Of course, we did not explore all mechanisms available to the cell. For instance, as discussed above, in this simplified cell model we could not assess the difference in information transfer between 'type 1' (or integrator) and 'type 2' (or resonator) or 'type 3' (or differentiator) cells, as this can only be modelled as a difference in the bifurcation from resting to spiking (a saddle-node versus a Hopf bifurcation). Moreover, we used the expIF model as a proof-of-principle of the effects of different intrinsic cell properties on information transfer and did not extensively fit the model to the experimental data. Indeed, the recorded spike trains are better classifiers than even the best-performing expIF model for the slow input current, suggesting that there are more relevant dynamic properties that are not captured by such a simplified model. However, using such a simple setup allows us to make several predictions that can be tested experimentally: we predicted that 1) blocking 'h-currents' will decrease the amount of information that is transferred 2) blocking subthreshold potassium currents will not have such an effect, and 3) there is an optimal range for the membrane conductance.

The heterogeneity of neuron properties has received much interest lately: for instance, it has been shown that heterogeneity in neural populations can increase coding robustness and efficiency [48], help optimize information transmission [80], increase network responsiveness [81], promote robust learning [82], help to control the dynamic repertoire of neural populations [83] and improve the performance on several tasks [84, 85]. Here, we show that in particular, the population of excitatory neurons of the barrel cortex shows a large variability in their intrinsic and response properties. Why the variability of the properties of excitatory neurons is larger than that of inhibitory ones is an exciting question, which is a subject for future experimental and computational evaluation. Moreover, the intrinsic properties of cortical neurons are under top-down influence by neuromodulators such as serotonin, acetylcholine and dopamine [86, 87]. Using the protocol described herein, it will be possible to investigate how these neuromodulators affect the intrinsic neural properties and, consequently, their information transfer. This will help reveal how the specific actions of these neuromodulators on the intrinsic properties of specific cell classes affects information transfer in the cortex. By investigating the relationship between intrinsic neuron properties and information transfer, we can begin to predict the effect of top-down processes on cortical processing.

## Supporting information

**S1 Fig. Threshold behaviour in the current clamp step-and-hold protocol.** A) Thresholds of all spikes during the step protocol. B) Thresholds of the first spikes after the step current initiation. IC Threshold adaptation: difference in threshold between the first and the last spike of the response. D) Last ISI length relative to the first ISI of the response. Excitatory (red) and inhibitory (blue) neurons. NB Results for significance testing in S1 Table.
(PDF)

**S2 Fig. Alternative calculation of mutual information.** A) Normalized histogram of all (black) input current, the input current when the hidden state was 1 (magenta) and when it was 0 (green) for an excitatory neuron. B) Histogram of the average membrane potential values over each state (i.e. the time between state switches) for all (black) states, the 1 states (magenta) and the 0 states (green) for an excitatory neuron. C) Same as B) but for the instantaneous firing rate (spike train convolved with an exponential kernel, see Materials and methods). D) Average I-V curve: each dot is the average of a single state for the input current (horizontal axis) and membrane potential (vertical axis). A linear curve was fitted for each analysis window to determine the slope of the $I - V$ curve, E) Same as D), but with the instantaneous firing rate on the vertical axis. F) The Mutual information between the hidden state and the instantaneous firing rate (black, left) and the Kullback-Leibler divergence $D_{KL}$ between the distributions of the instantaneous firing rates when the hidden states were 1 or 0 (magenta, right, see panel E) dependence on the number of bins used to bin the instantaneous firing rates (thin lines: single analysis window, lines with dots: average over windows). Based on this panel and panel L, the number of bins was set at 500. G-L) Same as A-F) but for inhibitory neurons. M) Comparison of the two methods to calculate the mutual information between the hidden state and the output spike trains (blue dots: inhibitory neurons, red dots: excitatory neurons). N) The Kullback-Leibler divergence between the histograms of the average over the state when the state was 1 or 0 (i.e. the 'separability') of the current (horizontal axis) and membrane potential (vertical axis). Note that for both inhibitory and excitatory neurons the membrane potential has a higher $D_{KL}$ than the input current. O) Same as N), but for the membrane potential and the instantaneous firing rate. P) The relation between the slope of the $I - V$ curve (see panels D and J) and the mutual information between the hidden state and the spike train. Note when comparing this to Fig 10 that here we depict the resistance $R = \frac{1}{g}$ rather than the conductance. One cell was excluded due to trial labelling issues.
(PDF)

**S1 Table. Statistical tests of the comparison between excitatory and inhibitory neurons in the current clamp step protocol.**
(DOCX)

**S2 Table. Statistical tests of the comparison between excitatory and inhibitory neurons in the frozen noise protocol (see main text Fig 5).** P-values were compared to a threshold of 5% / 6 groups = 29 0.83% (Bonferroni correction).
(DOCX)

**S3 Table. Statistical tests of the comparison between excitatory and inhibitory neurons receiving the control frozen noise stimulus (see main text Fig 5).** P-values were compared to a threshold 34 of 5% / 6 groups = 0.83% (Bonferroni correction).
(DOCX)

## Author Contributions

**Conceptualization:** Fleur Zeldenrust, Niccolò Calcini, Tansu Celikel.

**Data curation:** Fleur Zeldenrust, Niccolò Calcini, Tansu Celikel.

**Formal analysis:** Fleur Zeldenrust, Niccolò Calcini, Tansu Celikel.

**Funding acquisition:** Fleur Zeldenrust, Tansu Celikel.

**Investigation:** Niccolò Calcini, Xuan Yan, Ate Bijlsma.

**Methodology:** Fleur Zeldenrust, Niccolò Calcini, Tansu Celikel.

**Software:** Fleur Zeldenrust, Niccolò Calcini, Tansu Celikel.

**Supervision:** Fleur Zeldenrust, Tansu Celikel.

**Visualization:** Fleur Zeldenrust, Niccolò Calcini, Ate Bijlsma, Tansu Celikel.

**Writing – original draft:** Fleur Zeldenrust, Niccolò Calcini, Tansu Celikel.

**Writing – review & editing:** Fleur Zeldenrust, Niccolò Calcini, Tansu Celikel.

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
