## [Decision Letter · Decision Letter 0]

31 Dec 2023

Dear Dr Zeldenrust,

Thank you very much for submitting your manuscript "The tuning of tuning: how adaptation influences single cell information transfer" for consideration at PLOS Computational Biology.

As with all papers reviewed by the journal, your manuscript was reviewed by members of the editorial board and by several independent reviewers. In light of the reviews (below this email), we would like to invite the resubmission of a significantly-revised version that takes into account the reviewers' comments.

We do note that to fully address this feedback will likely require considerable effort.

We cannot make any decision about publication until we have seen the revised manuscript and your response to the reviewers' comments. Your revised manuscript is also likely to be sent to reviewers for further evaluation.

Sincerely,

Lyle J. Graham

Section Editor

PLOS Computational Biology

Reviewer's Responses to Questions

**Comments to the Authors:**

Reviewer #1: The paper describes a combination of modeling and ex-vitro experiments (in mouse barrel cortex) on excitatory and inhibitory neurons in mouse barrel cortex. The goal of the work is to elucidate the effect on information transmission of certain adaptive mechanisms in the neuron’s spike generating mechanism. This work relates to an important issue in neuroscience: How do individual neurons change some of their physiological properties to optimize signal transfer, as measured by Shannon information. The combination of actual experiments with detailed simulation is nice, since it allows the authors to explore a number of well-known properties of the cells that are not amenable to direct experimental manipulation. Both the analysis and the experiments are described in sufficient detail, and seem to be carried out very well.

Major comments

Overall, I do not have any issues with how the project was executed and with the results that were obtained in both experiment and simulation. I do, however, have a serious issue with the overall interpretation of the results in a wider context: Throughout the paper the authors quantify mutual information between a source (a 2-state Markov model) and a receiver (both the output of an Artificial Neural Network (ANN), or the spiking output of a neuron using the ANN signal as its input). The methods for analyzing this situation were worked out in reference 15, and they look correct to me. The authors claim correctly that this method is efficient, in the sense that one can get reliable estimates in relatively short time (order of 5 minutes), as compared to more exhaustive methods that may take an hour or more of experiment time. I can appreciate that it is crucial to use fast methods in an experiment where the cells cannot be held for very long.

However, there is an important downside to the analysis as well, which is nowhere discussed: That is that the source is likely to be an impoverished representation of the signals that are representative of the whisker system. The main information measures presented here quantify mutual information between the hidden Markov state on the one hand and the neuron’s firing rate on the other. In other words, what is measured is the relation between a two-state system and a spike count. Spike timing is not considered as an information carrier, and neither is the dynamic structure of more-or-less natural inputs. Over the last few decades, it has become clear that in many sensory systems, spike timing plays an important role in information transmission (indeed, many of the key publications are cited in the present paper). Over roughly that same time frame a large body of literature has developed which points to the whisker system, and neurons in barrel cortex as sparse encoders, which can be very finely tuned to temporal structure (e.g. the encoding of textures as sensed by the whiskers). None of that literature is cited here. So, while I understand the necessity of using a fast and efficient method for measuring information transmission, it is incumbent upon the authors to also discuss its limitations in the more general context of neural information transmission, in particular in the barrel cortex, but perhaps also in other modalities. They should, I think, also acknowledge that the low spike rate, and the low information throughput of the excitatory cells that they report might well be an artefact of their particular methods (case in point: because of the different firing rates of excitatory and inhibitory cells, the Markov switching rates are adjusted for each cell type: That already shows that the method lacks generality). Conceivably, for example, a full analysis might prove that the excitatory cells are highly selective to particular stimulus features, and may therefore signal a lot of information per spike. I do not mean to suggest that the authors perform a whole new set of (potentially impossible) experiments, but I do think it is important to address carefully the limitations of their particular analysis.

Minor points

Line 50: information processing rate -> information transmission rate?

Line 62: these to forms -> these two forms

Legend to Fig 1E: Contrary to the test in the legend, it looks to me that the red cloud of AP half widths of the excitatory cells overlaps completely with those of the inhibitory cells (and the shortest red ones are shorter than the shortest blue ones). But that is inconsistent with panel I of the same figure. Am I missing something?

Line 202: and, and -> and

Eq. 8: It is not clear how to interpret the smoothing parameter lambda: IN the equation it looks like a multiplicative factor, but that can’t be true. Does it represent the width of some convolution filter? Please explain.

Line 373: to entangle -> to disentangle

Fig 6: The right box contains a legend with valued of tau_adap: What is tau_adap, how is it related to the tau values in Eqs 14 and 15?

Fig 10: Should the units on the horizontal axes of panels C and D be milliseconds^-1 (that is, not seconds^-1 as stated). From the typical values of g_m and C_m one would get time constants of millisecond range

Line 666: What is “information compression” in this context?

Fig. 11: Panels A, C, E: the STA’s are significantly anti-causal. I assume that this is due to some smoothing steps somewhere in the chain of operations, but it also seems to contradict the statement that the stimuli are whitened. Text to panel B: Should “blue line” be replaced by “red line”?

Reviewer #2: This paper addresses an interesting question, the influence of different aspects of adaptation on the ability of neurons to transmit information about their inputs, with a comparison between different neuronal cell types as well as models. The approach here is one that I find rather convoluted and strongly influenced by a specific interpretation of neural firing advanced in previous work by members of this group. It is a very specific and not clearly correct version of information that only pertains to the ability to discriminate two mean input levels in the presence of noise, an ability that can be influenced by internal neuron dynamics in many more ways that are considered here. While I appreciate the significant work that has been put into this analysis, the difficulties in this logic create significant difficulty in interpreting these hardwon results.

The core of the approach is based on the influential and interesting work by Sophie Deneve and colleagues, including the first author of this paper, who postulated that neural systems may be performing a sort of inference on their inputs, and that their firing properties may be governed by the dynamics of this inference. As a toy problem to illustrate this idea, this previous work proposed a situation in which a system may be faced with signals that may come from one of two sources, each with a certain mean that is obscured by noise. In this case she derived equations for the time-varying estimate of the log-odds of the input being drawn from one or the other of these distributions and related the properties of the computations to biophysical properties of neurons. This idea that an underlying process of ongoing inference in the presence of noise is a mechanism of determining the timing of spikes is very creative: but it is simply an interpretation of the essential mechanism that drives a single neuron to fire. Rather, the timing of spikes of single neurons are governed by detailed temporal fluctuations in the time-varying current that they receive, with no intrinsic meaning needed to be ascribed to “noise” or “signal. For that reason, many previous approaches to quantifying information transmission in neural systems have used complex time-varying inputs and devised methods of computing information, including by giving repeated inputs to estimate noise entropy.

In this paper, however, an alternative method is used that takes its inspiration from the “Bayesian neuron” concept. Here one drives the neuron with an input with a “hidden state” that switches between two values, and reduces the evaluation of information to that of computing the information that the firing rate carries about the value of the hidden state, ie a maximum of one bit. This on its own is artificial, and a vast simplification of neuronal response, but OK. The signal is generated as a sum of 1000 Poisson processes for which the hidden state serves as the rate; each point process is then convolved with an exponential kernel and summed. Thus, the time-varying hidden state is transformed into something close to a white noise process with a local mean and standard deviation. One can then ask how well can one determine the mean from a chunk of signal in the presence of this noise; and then, how does driving a spiking neuron with this input, re-representing it in spiking output, affect this level of discriminability. While in general one could have considered discriminating two mean values of rate, here the readout is taken to be spikes or no spikes, ie requiring the system to be in a regime in which the DC level of the lower state is subthreshold for spiking. Because of this, the information should be limited by several things. The most important effect should be the nature of the f-I curve (firing rate vs mean current) for the neuron: how steep it is, how far apart are the input DC values, governing how well separated for these chosen values the mean firing rate of the neuron will be, how the firing rate as a function of the mean current is influenced by background noise (Chance and Abbott; Higgs and Spain), how the variance of the effective background noise changes this relationship. One should also discuss the choice of the time bin in which to probe the firing, and how this interacts with the neuron’s refractory period. None of these issues are explicitly discussed.

The issue in which the paper invests most analytical machinery is that of the time dependence of the switches. This is important as it will interact with the temporal integration time of the neuron. In general, it will take some time following a switch in the input DC before the steady state firing rate for the new input is reached, which can be due to the neuron’s intrinsic filtering properties, or to internal dynamics of the neuron that change in response to the change in DC input. Thus, the ability to discriminate a change will depend on how rapidly the hidden state is changing. If the state varies so rapidly that the receiving neuron does not reach steady state, it will be difficult to make the necessary discrimination. Thus generally over some fraction of the time in which the input is held at a given hidden state, and again depending on the nature of the processing of that neuron, the ability to discriminate the DC level will be nonoptimal.

The information calculation is done in a way that to me was confusing and is very reliant on the “Bayesian neuron” concept. To evaluate the information in the input about the hidden state, one could simply form the distributions of input (averaged, presumably, over bins of a chosen size) conditioned on times at which the state was high and in which it was low, and compute the mutual information as the entropy of the total input distribution minus the appropriately weighted entropy of the conditional distributions. Given the simple nature of the input, this mutual information could probably be computed analytically. One would then do the same for the firing rate. This would give the MI between state and input, and the MI between spikes and input, from which one could compute the fraction. The rate of the switches between high and low will generally always matter to this information calculation because of the neuron’s multiple time-constant filtering and processing properties (whether it is intrinsically an integrator or differentiator, whether it is fast-filtering or slow, whether there are slow time constants of adaptation that depend on the rate of switching), putting different fractions of the firing rate into a transient “ambiguous” zone depending on this switching rate, so I’d be inclined to plot the information out as a function of that switching time.

As an aside-- it’s not obvious that a spiking neuron necessarily loses information about this mean relative to the input itself. For example, as alluded to above, a neuron that acts as a fractional differentiator in a range of timescales that are close to the switching rate (ie is less sensitive to the noise but rather responds to the DC) could even highlight the moment of state transition relative to the unfiltered original input, as well as retain information about the steady state (Lundstrom et al, 2007).

Overall the approach taken in the paper relative to the simpler version outlined above seems like an overly model-driven approach, in which clarity about the role of different mechanisms was lost.

A few specific comments.

It seems an exaggeration to call the stimulus construction procedure an “ANN”. It is simply the sum of many independent Poisson processes, filtered with synaptic time course. Calling it an ANN implies that there is connectivity or meaningful dynamics generated by these simulated neurons.

It is unclear and not explained why the input has to be “frozen” noise. Is this to hold the relative times spent in the two states and in the transitions betwee

---

## [Decision Letter · Decision Letter 1]

1 Apr 2024

Dear Dr Zeldenrust,

We are pleased to inform you that your manuscript 'The tuning of tuning: how adaptation influences single cell information transfer' has been provisionally accepted for publication in PLOS Computational Biology.

Best regards,

Lyle Graham

Section Editor

PLOS Computational Biology

Reviewer's Responses to Questions

**Comments to the Authors:**

Reviewer #1: I commend the authors on their response and the rewriting of crucial aspects of the paper. In particular the “conclusion and discussion” section is much improved, discusses some crucial limitations of the methodology, and puts the experimental/computational results and their interpretation in proper perspective. They have also properly addressed the minor points that I raised earlier.

Typos:

Line 166: Rewrite “This had of 2 reasons”

Legend Fig 6: Correct the TeX code in “The time constant of the adaptation, τtextrmadap, corresponds to τw for subthreshold adaptation”.

Reviewer #2: The authors have addressed the critiques.

**Have the authors made all data and (if applicable) computational code underlying the findings in their manuscript fully available?**

Reviewer #1: Yes

Reviewer #2: None

PLOS authors have the option to publish the peer review history of their article (what does this mean?). If published, this will include your full peer review and any attached files.

Reviewer #1: No

Reviewer #2: No

---

## [Editor Report · Acceptance letter]

3 May 2024

PCOMPBIOL-D-23-01643R1 

The tuning of tuning: how adaptation influences single cell information transfer

Dear Dr Zeldenrust,

I am pleased to inform you that your manuscript has been formally accepted for publication in PLOS Computational Biology. Your manuscript is now with our production department and you will be notified of the publication date in due course.

With kind regards,

Anita Estes
